# The APPL1-Rab5 axis restricts NLRP3 inflammasome activation through early endosomal-dependent mitophagy in macrophages

Kelvin Ka Lok Wu[1], KeKao Long[1], Huige Lin[1], Parco Ming Fai Siu [2], Ruby Lai Chong Hoo[3,4], Dewei Ye[5], Aimin Xu [3,4,6] & Kenneth King Yip Cheng [1✉]

Although mitophagy is known to restrict NLRP3 inflammasome activation, the underlying regulatory mechanism remains poorly characterized. Here we describe a type of early endosome-dependent mitophagy that limits NLRP3 inflammasome activation. Deletion of the endosomal adaptor protein APPL1 impairs mitophagy, leading to accumulation of damaged mitochondria producing reactive oxygen species (ROS) and oxidized cytosolic mitochondrial DNA, which in turn trigger NLRP3 inflammasome overactivation in macrophages. NLRP3 agonist causes APPL1 to translocate from early endosomes to mitochondria, where it interacts with Rab5 to facilitate endosomal-mediated mitophagy. Mice deficient for APPL1 specifically in hematopoietic cell are more sensitive to endotoxin-induced sepsis, obesity-induced inflammation and glucose dysregulation. These are associated with increased expression of systemic interleukin-1β, a major product of NLRP3 inflammasome activation. Our findings indicate that the early endosomal machinery is essential to repress NLRP3 inflammasome hyperactivation by promoting mitophagy in macrophages.

[1] Department of Health Technology and Informatics, The Hong Kong Polytechnic University, Hung Hom, Hong Kong SAR, China. [2] School of Public Health, The University of Hong Kong, Pok Fu Lam, Hong Kong SAR, China. [3] The State Key Laboratory of Pharmaceutical Biotechnology, The University of Hong Kong, Pok Fu Lam, Hong Kong SAR, China. [4] Department of Pharmacology & Pharmacy, The University of Hong Kong, Pok Fu Lam, Hong Kong SAR, China. [5] Guangdong Research Center of Metabolic Diseases of Integrated Western and Chinese Medicine, Guangdong Pharmaceutical University, Guangzhou, China. [6] Department of Medicine, The University of Hong Kong, Pok Fu Lam, Hong Kong SAR, China. ✉email: kenneth.ky.cheng@polyu.edu.hk

Activation of nucleotide binding domain leucine-rich repeat-containing receptor, pyrin domain-containing-3 (NLRP3) inflammasome is triggered by infection and cellular stress encountered by the sensors including pathogen-associated or damage-associated molecular patterns (PAMPs and DAMPs, respectively). In response to PAMPs and DAMPs, NLRP3 undergoes oligomerization and binds with the adaptor apoptosis-associated speck-like protein (ASC) through their pyrin domain. Pro-caspase-1 is then recruited to the caspase activation and recruitment domain (CARD) on ASC, which causes auto-catalytic cleavage of pro-caspase-1 to active form caspase-1 to induce subsequent maturation and secretion of interleukin (IL)-1β and IL-18 in immune cells[1]. Transient activation of this highly coordinated immune response removes foreign pathogens and cellular debris and hence restores tissue homeostasis, but its persistent and unresolved activation leads to autoimmune disease, sepsis as well as cardiometabolic diseases[2–5].

Emerging evidence indicate that danger signals including reactive oxygen species (ROS) and oxidized mitochondrial DNA (mtDNA) from damaged mitochondria elicit NLRP3 inflammasome activation[6]. The increase of mitochondrial ROS (mtROS) induces cysteine oxidation of cellular proteins and oxidizes cytosolic mtDNA, both of which activate NLRP3 inflammasome[7,8]. In addition, NLRP3 inflammasome and the adaptor ASC are recruited and assembled onto mitochondria by the regulation of the mitochondrial adaptor MAVS and Sirtuin2-acetylated α-tubulin pathways[9,10]. On the contrary, elimination of mtROS or cytosolic mtDNA or inhibition of mtDNA synthesis abolishes NLRP3 inflammasome overactivation[11–14]. Mitophagy is a specific form of autophagy, which removes damaged mitochondria via lysosomes for maintaining mitochondrial quality and cell homeostasis[6,11–13,15]. In response to mitochondrial depolarization, the mitochondrial associated kinase PINK1 (the PTEN-induced kinase 1) phosphorylates the E3 ubiquitin ligase Parkin on the mitochondria, which in turn triggers polyubiquitination and subsequent recruitment of autophagy receptors (such as p62) to the mitochondria[1,16–18]. Apart from lysosome, several organelles including proteasome, endoplasmic reticulum and endosomes have been shown to participate in mitophagic responses at multiple sites and steps[19–23]. In particular, the endosomal proteins such as the early endosomal protein Rab5, the late endosome associated protein Rab7, CHMP2A, and CHMP4B involve in different steps of mitophagy via Parkin-dependent and -independent manners, which include autophagosome formation, targeting damaged mitochondria to lysosome, and encapsulation of the damaged mitochondria by autophagic membrane as well as phagophore sealing[19,20,23,24]. These findings indicate the complexity and interconnectivity of mitophagic pathways.

The adaptor protein containing NH2-terminal Bin/Amphiphysin/Rvs (BAR) domain, a central pleckstrin homology (PH) domain and a COOH-terminal phosphotyrosine-binding domain (PTB) 1 (APPL1) is an early endosome resident protein with ability to mediate the crosstalk between multiple signaling pathways on the endosomes. In response to cellular stress and hormonal factors, APPL1 translocates among different intracellular organelles including early endosomes, plasma membrane and nucleus, and controls apoptosis, glucose metabolism, and proliferation[25–27]. In this study, we showed that APPL1 and its interacting partner GTPase Rab5 promote early endosome-mediated elimination of damaged mitochondria, which in turn limits NLRP3 inflammasome activation in macrophages. In vivo, we demonstrated that hematopoietic APPL1 is crucial to restrict NLRP3 inflammasome activation in obesity and acute septic response. Our study uncovers a pathway by which early endosome-mediated mitophagy prevents NLRP3 overactivation in macrophages.

## Results

### Lack of APPL1 potentiates NLRP3 inflammasome activation in macrophages.
Although APPL1 has been shown to regulate lipopolysaccharide (LPS)-induced pro-inflammatory program in RAW264.7 cells[28] (a mouse macrophage cell line lacking functional inflammasome machinery[29]), its role in regulating inflammasome and M1 polarization has never been explored. To address this, bone-marrow-derived macrophages (BMDM) from APPL1 knockout (KO) mice and their wild-type (WT) littermates were treated with LPS and IFN-γ or primed with LPS followed by ATP stimulation, to induce M1 polarization or NLRP3 inflammasome activation, respectively. Immunoassay showed that the cytokines associated with M1 macrophage activation (which are tumor necrosis factor alpha [TNF-α], monocyte chemoattractant protein-1 [MCP-1] and IL-1β) increased to a similar extent in both APPL1 KO and WT BMDM after stimulation with the M1-polarizing agent (Fig. 1a). Consistent to our data, two studies also showed that knockdown of APPL1 expression does not alter LPS-induced production of TNF-α in macrophages[28,30]. Treatment with the NLRP3 inflammasome activator ATP further provoked mature IL-1β secretion but had no effect on MCP-1 and TNF-α secretion in WT BMDM (Fig. 1a), whereas genetic deletion of APPL1 selectively augmented the effect of ATP on IL-1β secretion (Fig. 1a). LPS + ATP-induced secretion of IL-18 (another inflammasome-associated cytokine) in APPL1 KO BMDM was approximately 2-fold higher than that in WT controls (Fig. 1a). In addition, the potentiating effect of APPL1 deficiency on ATP-induced IL-1β secretion was dose- and time-dependent (Fig. 1b, c). Likewise, APPL1 KO BMDM displayed an augmented IL-1β production but normal TNF-α secretion upon stimulation with nigericin and monosodium urate microcrystals (both are the NLRP3 inflammasome activators), when compared with WT controls (Supplementary Fig. 1a, b). APPL1 deficiency did not affect IL-1β production in response to flagellin (an inducer of NLRC4), synthetic B-form double-stranded Poly(dA:dT) (which utilizes AIM2) or transfection of LPS (which utilizes noncanonical caspase-11 pathway) and had no obvious effect on pyroptosis (Supplementary Fig. 1c–g) and macrophage differentiation (Supplementary Fig. 2a). These findings indicate that the effect of APPL1 deficiency appears to be specific to NLRP3 inflammasome.

### APPL1 deficiency activates caspase-1-dependent IL-1β secretion.
The activation of inflammasome requires two signals; the first signal induces mRNA expression of pro-IL-1β and NLRP3 via NF-κB and the second signal causes autocatalytic cleavage of pro-caspase-1 to caspase-1 and subsequent maturation of IL-1β[31]. First, we examined LPS-induced NF-κB activation (the first signal) by measuring degradation of NF-κB inhibitor α (IκBα) and subsequent phosphorylation of p65. Immunoblotting analysis indicated that there was no difference in LPS-induced NF-κB activation between APPL1 KO and WT BMDM (Supplementary Fig. 2b). Consistently, mRNA and protein levels of NLRP3, pro-caspase-1 and pro-IL-1β were similar between the two groups (Fig. 1d and Supplementary Fig. 2c). On the other hand, expression of cleaved caspase-1 and its enzymatic activity in the culture medium isolated from APPL1-deficient BMDM with LPS + ATP stimulation were significantly increased compared to those from WT controls (Fig. 1e, f). Selective inhibition of caspase-1 activity with Ac-YVAD-cmk abolished APPL1 deficiency-induced over-secretion of IL-1β (Fig. 1g). Together, the data collectively suggested that APPL1 appears to control the second signal but not the first signal for NLRP3 inflammasome activation.

### APPL1 deficiency disrupts mitochondrial homeostasis in macrophages.
NLRP3 inflammasome activation is known to

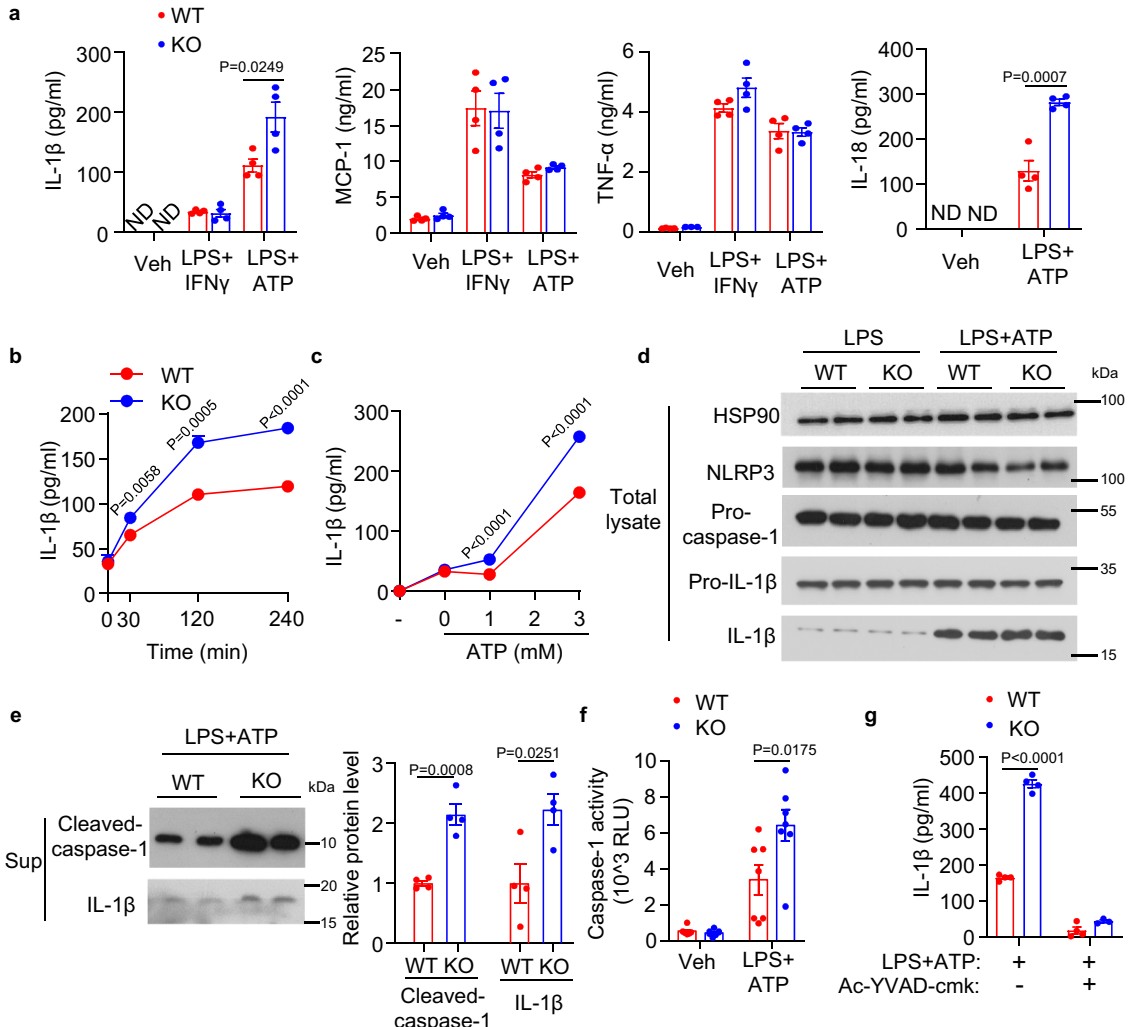

**Fig. 1 APPL1 deficiency promotes inflammasome activation via caspase-1 in bone-marrow-derived macrophages.** Bone-marrow-derived macrophages (BMDM) were isolated from 10–12-week-old APPL1 KO mice and their wild-type (WT) controls. To induce NLRP3 inflammasome activation, BMDM were primed with LPS (100 ng/ml) for 20 h, followed by treatment with ATP (3 mM) for 4 h. To induce M1 macrophage polarization, BMDM were treated with LPS and IFNγ (100 ng/ml for each) for 24 h. **a** Levels of IL-1β, MCP-1, TNF-α, and IL-18 in cell culture supernatant were measured by immunoassays. $n = 4$ biologically independent samples. **b** Time- and (**c**) dose-dependent effects of ATP on IL-1β secretion in LPS-primed BMDM. $n = 4$ biologically independent samples. **d** Cell lysate and (**e**) cell culture supernatant collected from LPS-primed BMDM with or without ATP stimulation were subjected to immunoblotting analysis using antibody against NLRP3, caspase-1, IL-1β, and HSP90. **e** The right panel is densitometry analysis of cleaved caspase-1 and IL-1β. $n = 4$ biologically independent samples. The samples of immunoblotting analysis were collected from the same experiment and blots were processed in parallel. **f** Caspase-1 activity in the supernatant was measured using Caspase-Glo® 1 Inflammasome Assay. $n = 7$ biologically independent samples. **g** The BMDM primed with LPS were pretreated with or without the caspase-1 inhibitor (Ac-YVAD-cmk) for 2 h before ATP stimulation for 4 h, followed by measurement of IL-1β in the cell culture supernatant. $n = 4$ biologically independent samples. ND (undetectable). Data are displayed as mean ± SEM. Statistical significance was tested using two-tailed student's *t*-test (**a**, **b**, **c**, **e**, **g**) or Mann–Whitney U test (**f**).

disrupt mitochondrial integrity, promote mtROS production and induce mitochondrial DNA (mtDNA) release and oxidation[12,13]. Indeed, APPL1 ablation led to a higher mitochondrial superoxide generation under basal or LPS and ATP stimulation (Fig. 2a). Using fluorescent TMRE assay, we showed that APPL1 deficiency resulted in a loss of mitochondrial membrane potential (Fig. 2b). Next we measured percentage of damaged and live mitochondria using MitoTracker Green (which stains mitochondria regardless of its membrane potential) and MitoTracker Deep Red (which stains mitochondria dependent on its membrane potential), respectively. Under basal condition, APPL1 KO BMDM displayed an approximately one-fold increase in percentage of damaged mitochondria than in WT BMDM, and such difference was also observed upon LPS + ATP stimulation (Fig. 2c). In addition, the amount of mtDNA and its oxidized form in the cytosol of APPL1

KO BMDM was much higher than that in WT BMDM after treatment with LPS and ATP but not under basal condition (Figs. 2d–e and 3f–g). On the other hand, mitochondrial DNA copy number and expression of mitochondrial proteins (including Tom20, cytochrome C [Cyto C] and pyruvate dehydrogenase) were similar between APPL1 KO and WT BMDM under LPS priming condition, indicating that APPL1 deficiency had no obvious impact on mitochondrial turnover under LPS alone condition (Supplementary 2D, E). Apart from mitochondrial damages, ATP is known to induce inflammasome activation via multiple pathways such as disrupting intracellular Ca$^{2+}$ and K$^{+}$ balance. To further investigate whether APPL1 specifically modulates mitochondrial stress-induced inflammasome activation, we treated LPS-primed APPL1 KO BMDM with the mitochondrial complex I inhibitor rotenone, the complex III inhibitor

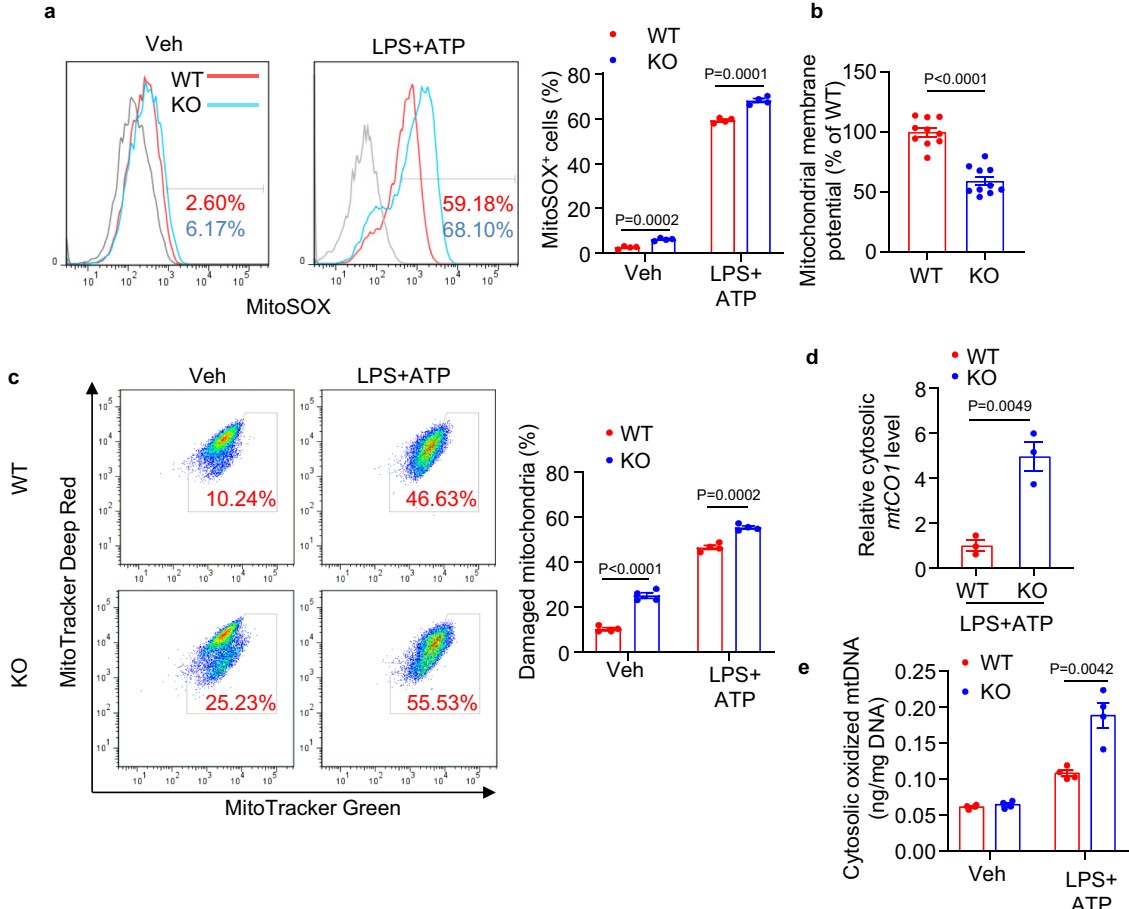

**Fig. 2 APPL1 deficiency leads to mitochondrial damage.** BMDM from APPL1 KO mice and their WT controls were primed with LPS (100 ng/ml) for 20 h, followed by stimulation with ATP (3 mM) for 30 min. **a** The treated BMDM were stained with MitoSOX™ Red Mitochondrial Superoxide Indicator, followed by flow cytometry analysis. Representative histogram plots of MitoSOX intensity in BMDM are shown. The right panel is the percentage of MitoSOX+ BMDM. $n = 4$ biologically independent samples. **b** Mitochondrial membrane potential determined by TMRE assay under the basal condition. $n = 10$ biologically independent samples. **c** Flow cytometry analysis of mitochondrial status in BMDM stained with MitoTracker Green and MitoTracker Deep red. The percentage represents BMDM with damaged mitochondria. $n = 4$ biologically independent samples. **d** Cytosolic mitochondrial DNA (mtDNA) was extracted from BMDM after ATP treatment, followed by QPCR analysis of *mitochondrial cytochrome c oxidase 1 (mtCO1)* level with normalization of nuclear DNA encoding *18 S ribosomal RNA*. $n = 3$. biologically independent samples. **e** 8-OH-dG level in the cytosolic mtDNA. $n = 4$ biologically independent samples. Data are displayed as mean ± SEM. Statistical significance was tested using two-tailed student's *t*-test (**a**–**e**).

antimycin A and respiratory chain uncoupler Carbonyl cyanide-p-trifluoromethoxyphenylhydrazone (FCCP), which have been shown to induce NLRP3 activation[13]. As expected, these mitochondrial-targeted compounds augmented IL-1β secretion in WT BMDM primed with LPS, and such a pro-inflammatory effect was more pronounced in APPL1-deficient BMDM (Fig. 4a). Under the same stimulation, APPL1 KO BMDM exhibited higher caspase-1 activity in culture medium but had similar TNF-α secretion, when compared to their WT controls (Fig. 4b, c). The degree of mitochondrial membrane potential loss induced by rotenone and antimycin A but not FCCP was significantly larger in APPL1 KO BMDM than that in WT controls (Fig. 4d).

Next, we investigated whether inhibition of the mitochondrial danger signals (i.e., mtROS and oxidized mtDNA) is able to alleviate elevated inflammasome activation in APPL1-deficient BMDM. Treatment with the general ROS scavenger N-acetyl cysteine (NAC) significantly reduced LPS + ATP-induced mtROS production in both APPL1 KO and WT BMDM, accompanied by a reduction of IL-1β and cleaved caspase-1 secretion in culture medium (Fig. 3a, c–e). Likewise, treatment with the mitochondrial-specific ROS scavenger MitoTEMPO reversed the elevated levels of mtROS, caspase-1, and IL-1β induced by

APPL1 deficiency (Fig. 3a, c–e). Noticeably, the increased number of damaged mitochondria in APPL1 KO BMDM could not be completely reversed by treatment with NAC nor MitoTEMPO, indicating mitochondrial damage as an upstream event relative to mtROS production (Fig. 3b). We speculated that the increased cytosolic oxidized mtDNA also contributes to the elevated NLRP3 inflammasome activation in APPL1 KO BMDM. To test this, we treated BMDM with a low dose of ethidium bromide to remove mtDNA as previously described[12]. QPCR analysis of the mitochondrial gene *cytochrome c oxidase 1 (mtCO1)* confirmed elimination of mtDNA in WT and APPL1 KO BMDM after ethidium bromide treatment for 4 days (Fig. 3f). Ethidium bromide treatment also reduced amount of oxidized cytosolic mtDNA in both WT and APPL1 KO BMDM under LPS + ATP stimulation (Fig. 3g). Elimination of oxidized mtDNA abolished APPL1 deficiency-induced elevation of IL-1β secretion (Fig. 3h). Indeed, inhibiting the binding of oxidized mtDNA to NLRP3 using 8-hydroxy-guanosine[8] (8-OH-dG) completely abolished APPL1 deficiency-induced inflammasome activation in BMDM under LPS + ATP condition (Fig. 3i, j). These data collectively suggest that APPL1 deficiency induces mitochondrial damage, leading to increased production of mtROS and cytosolic oxidized

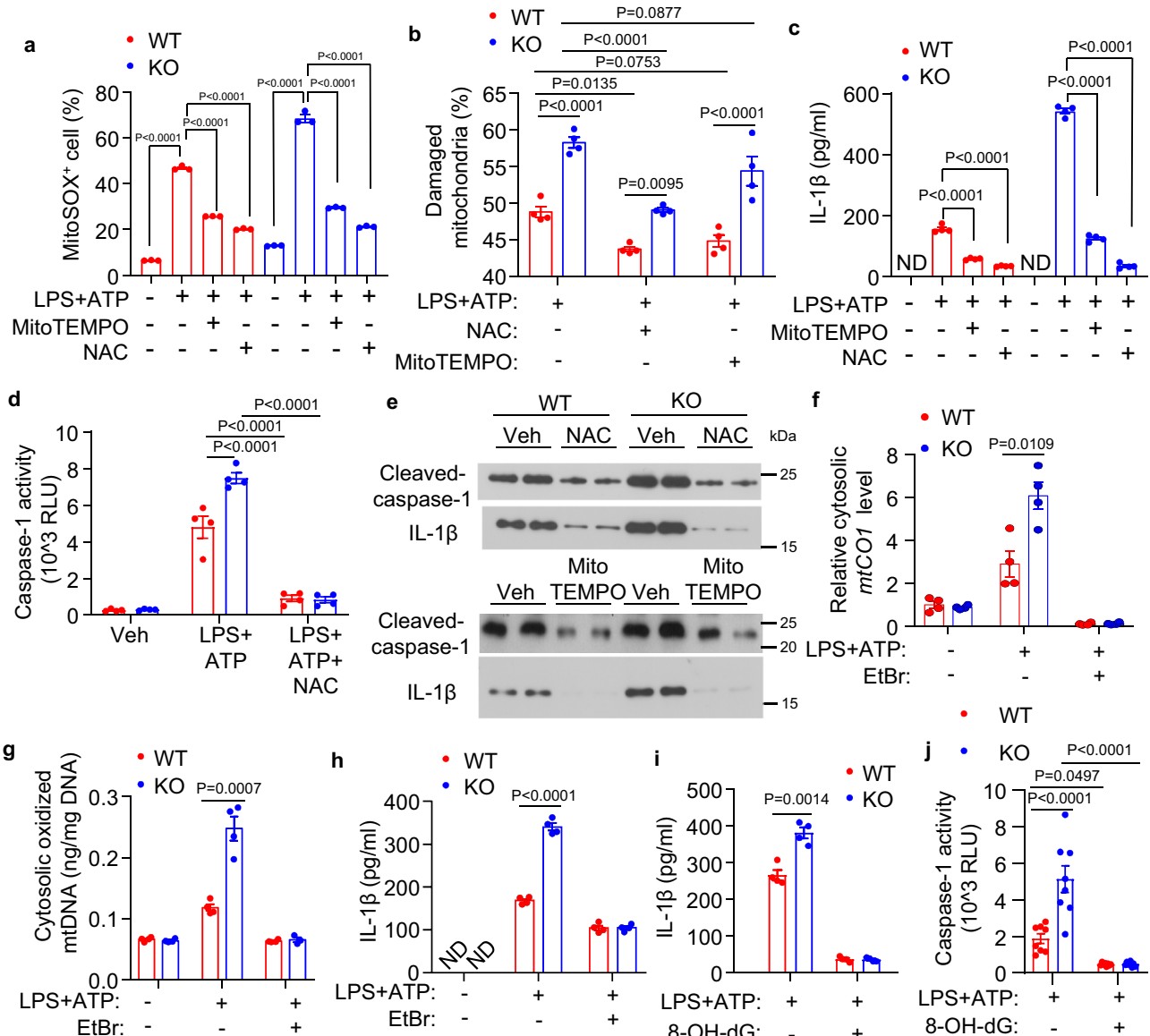

**Fig. 3 Removal of danger signals from mitochondria reverses excessive inflammasome activation in APPL1 *null* BMDM.** LPS-primed BMDM from APPL1 KO mice and their WT controls were treated with the ROS scavenger N-Acetyl Cysteine (NAC) or MitoTEMPO for 4 h and 30 min, respectively, followed by stimulation with ATP for 4 h. **a** Intracellular level of mitochondrial ROS (mtROS) was measured by MitoSOX assay as described in Fig. 2a. $n = 3$ biologically independent samples. **b** Percentage of damaged mitochondria in LPS-primed BMDM with ATP stimulation, which was determined by flow cytometry as described in Fig. 2c. $n = 4$ biologically independent samples. **c** IL-1β level in the cell culture supernatant. $n = 4$ biologically independent samples. **d** Caspase-1 activity in the cell culture supernatant was measured using Caspase-Glo® 1 Inflammasome Assay. $n = 4$ biologically independent samples. **e** Immunoblotting analysis of IL-1β and caspase-1 in the supernatant. $n = 4$ biologically independent samples. **f–h** LPS-primed BMDM were treated with ethidium bromide (EtBr; 100 ng/ml) for 4 days, followed by ATP stimulation for 4 h. $n = 4$ biologically independent samples. **f** Relative amount of cytosolic mtDNA (reflected by *mtCO1* level) determined by QPCR analysis as Fig. 2d. **g** 8-OH-dG level in the cytosolic mtDNA. $n = 4$ biologically independent samples. **h** IL-1β in the cell culture supernatant. $n = 4$ biologically independent samples. **i, j** LPS-primed BMDM were pretreated with or without 8-OH-dG (200 μM) for 3 h, followed by ATP stimulation for 4 h. **i** IL-1β and (**j**) caspase-1 activity in the cell culture supernatant. Panel **i**: $n = 4$; Panel **j** $n = 8$ biologically independent samples. ND (Undetectable). Data are displayed as mean ± SEM. Statistical significance was tested using two-tailed student's *t*-test (**f–i**) or one-way ANOVA with post-hoc Bonferroni correction (**a, b, c, d, j**).

mtDNA, which in turn promotes NLRP3 inflammasome activation and IL-1β production.

The excessive accumulation of cytosolic mtDNA might induce inflammasome overactivation in APPL1 KO BMDM, via AIM2 or cGAS-STING-dependent pathway[32–34]. To test the above possibility, we employed siRNA to knockdown *AIM2* or *STING* expression in APPL1 KO BMDM. As expected, downregulation of AIM2 or STING abolished Poly (dA:dT)-induced IL-1β production in both APPL1 KO BMDM or WT BMDM, yet it

had minimal effect on APPL1 deficiency-induced augmented NLRP3 inflammasome activation under LPS + ATP or LPS + FCCP condition (Supplementary Fig. 3a–e). Surprisingly, pharmacological inhibition of cGAS using RU521 partially attenuated APPL1 deficiency-induced NLRP3 inflammasome activation (Supplementary Fig. 3f). However, comparing with the inhibitory effects observed in MitoTEMPO, ethidium bromide and 8-OH-dG experiments (Fig. 3), IL-1β production remained significantly higher in APPL1 KO BMDM than that in

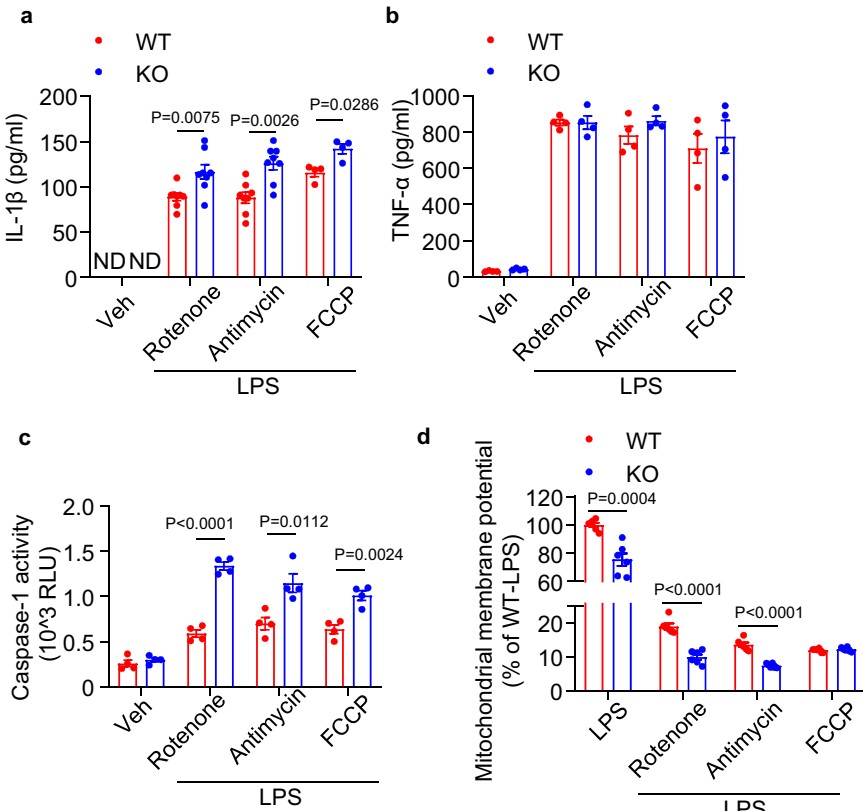

**Fig. 4 APPL1 controls inflammasome activation induced by mitochondrial inhibitors or uncoupler.** BMDM from APPL1 KO mice and their WT controls were primed with LPS (100 ng/ml) for 20 h, followed by treatment with the mitochondrial inhibitors (rotenone [10 μM] or antimycin A[40 μg/ml]) or the uncoupler carbonyl cyanide-4-(trifluoromethoxy) phenylhydrazone (FCCP, 25 μM]) for 6 h. The BMDM without any stimulation is denoted as vehicle control (Veh). **a** IL-1β and (**b**) TNF-α levels in the cell culture medium were measured by the immunoassays. $n = 4$ biologically independent samples. **c** Caspase-1 activity in the cell culture medium was measured using Caspase-Glo® 1 Inflammasome Assay. $n = 4$ biologically independent samples. **d** Mitochondrial membrane potential was measured by TMRE assay. The data are presented as percentage of WT-LPS. $n = 6$ biologically independent samples. Data are displayed as mean ± SEM. Statistical significance was tested using two-tailed student's $t$-test (**b**–**d**) or Mann–Whitney U test (**a**).

WT BMDM under RU521 treatment condition. These data collectively suggest that AIM2 or cGAS/STING might not be the major mediator of APPL1 deficiency-induced NLRP3 inflammasome activation in BMDM, and confirmation of these results using AIM2/STING knockout cells in future studies is warranted.

**APPL1 deficiency abrogates the trafficking of mitophagosome to lysosome for degradation.** Excessive accumulation of damaged mitochondria in APPL1 *null* BMDM indicate that APPL1 deficiency might lead to defective mitophagy. To test this, we employed multiple approaches to assess mitophagy response. First, we measured dynamic expressions of the mitochondrial proteins Tom20 and Cyto C in APPL1 KO and WT BMDM. Tom20 is a mitochondrial import receptor subunit that localizes on the outer mitochondrial membrane and Cyto C is a mitochondrial matrix protein, and hence they are commonly used as mitochondria markers to reflect mitochondrial clearance and mitophagic rate[16,17,23]. Immunoblotting analysis revealed that Tom20 and Cyto C expression gradually decreased (which also indicates decreased of mitochondrial mass) in LPS-primed WT BMDM upon stimulation with ATP (Fig. 5a). Under the same stimulation, the degradation of Tom20 and Cyto C were markedly abolished by APPL1 deficiency (Fig. 5a). Since we observed that APPL1 deficiency also promoted FCCP-induced inflammasome activation and FCCP is known to regulate mitophagy via PINK1-parkin pathway[16,17], we hypothesized that APPL1 also controls FCCP-induced mitophagy. Indeed, FCCP-mediated

Tom20 and Cyto C degradation were substantially abrogated by APPL1 deficiency (Fig. 6a). Second, we used the mitophagy reporter Mito-Keima to quantitatively measure mitophagic activity[35,36]. WT and APPL1 KO BMDM were infected with adenovirus encoding Mito-Keima for 24 h, followed by NLRP3 inflammasome activation as above. Flow cytometry analysis showed that WT and APPL1 KO BMDM exhibited similar mitophagy at baseline level, as reflected by similar percentage of the cells with a high 561/405 nM ratio (Figs. 5b, 6b and Supplementary Fig. 4). As expected, FCCP or ATP markedly increased the percentage of cells with the high 561/405 nm ratio in WT BMDM (from 6.154 to 16.757% and 10.097 to 20.363%, respectively), whereas APPL1 deficiency diminished these mitophagic responses (from 6.377 to 10.694% and 9.594 to 14.125%) (Figs. 5b and 6b). Third, we monitored mitophagy by co-staining mitochondria (labelled by MitoTracker Deep Red) with LC3 (an autophagy marker) or with LAMP1 (a lysosome marker) to reflect formation of mitophagosomes and mitochondrial autolysosomes, respectively. Confocal images revealed that ATP stimulation induced a trend of increased colocalization of MitoTracker with LC3+ puncta in LPS-primed WT BMDM (Fig. 5c). Whereas APPL1 KO BMDM displayed a significant increased colocalization of LC3+ puncta and MitoTracker under LPS + ATP stimulated condition (Fig. 5c), indicating that the mitochondria inside autophagosome is probably not eliminated by the autophagosomal machinery. We examined whether defective mitophagic flux contributes to the accumulation of damaged mitochondria and

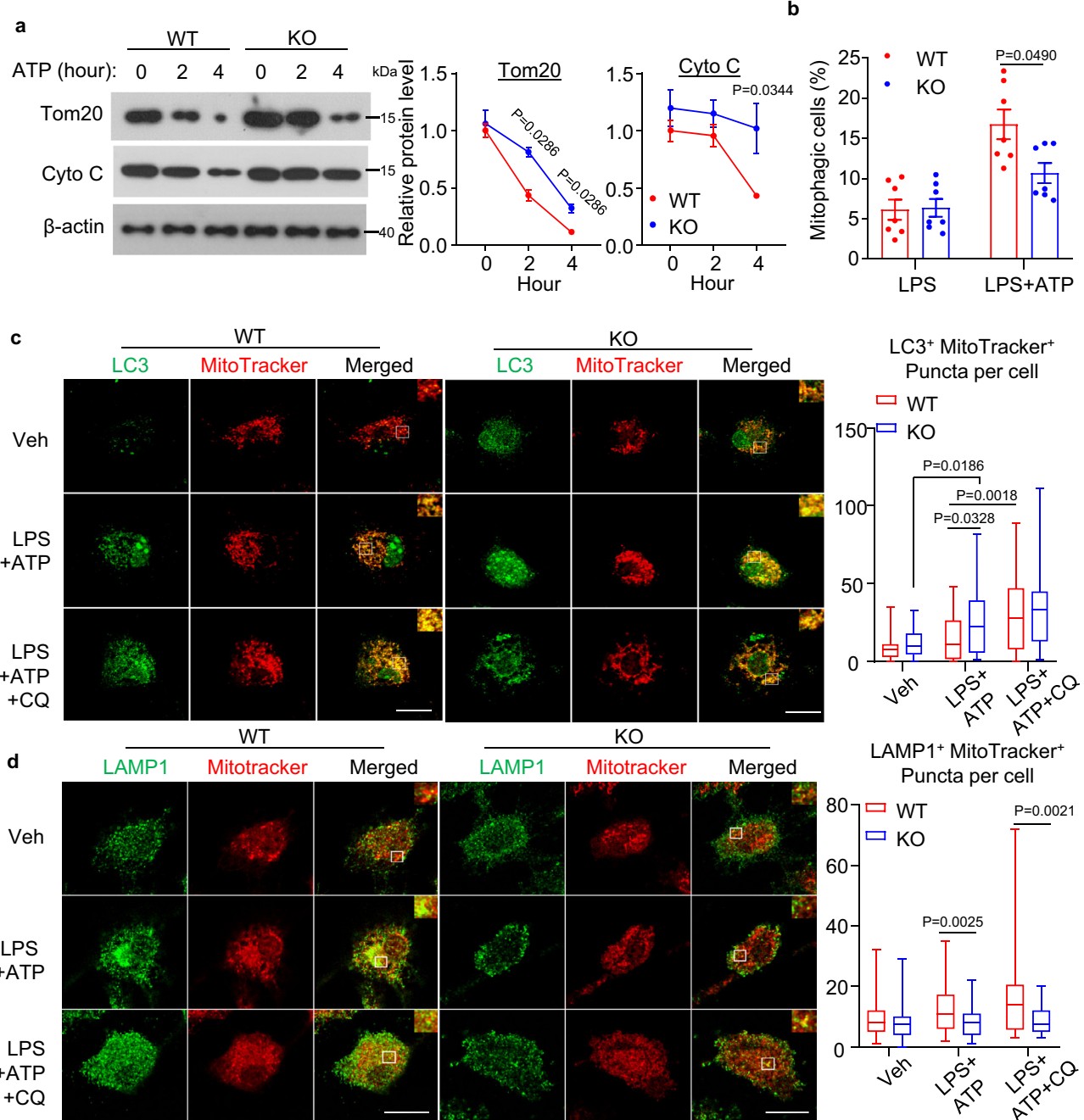

**Fig. 5 Ablation of APPL1 impairs mitophagy in macrophages.** BMDM from APPL1 KO mice and their WT controls were primed with LPS for 20 h, followed by stimulation with ATP as indicated. **a** The cell lysates were subjected to immunoblotting analysis of the mitochondrial proteins Tom20 and cytochrome c (Cyto C) and the loading control β-actin. The line charts are densitometry analysis of Tom20 and Cyto C normalized with β-actin. $n = 4$ biologically independent samples. The samples of immunoblotting analysis were collected from the same experiment and blots were processed in parallel. **b** Percentage of mitophagic cells after treatment with ATP for 30 min was determined by cytometry analysis using Mito-Keima. The gating strategy for identification and quantification of mitophagic cells and representative flow cytometry plots are included in Supplementary Fig. 4. $n = 7$ biologically independent samples. **c**, **d** LPS-primed BMDM were treated with chloroquine (CQ, 20 μM) for 1 h, followed by stimulation with ATP for another 1 h. The BMDM were subjected to immunofluorescence staining of **c** LC3 (green) and MitoTracker Deep red (red) or (**d**) LAMP1 (Green) and MitoTracker Deep Red (red). The boxplots on the right panels are quantification of colocalization of LC3$^+$ puncta with MitoTracker (**c**) and colocalization of LAMP1 and MitoTracker (**d**). $n = 50$ biologically independent cells. Scale bar: 10 μm. Representative images were shown. Data are displayed as mean ± SEM. In box and whisker plots, the whiskers extend to the minimum and maximum values, while the box presents the 25th and 75th percentiles with the central line at the median. Statistical significance was tested using two-tailed student's $t$-test (**a** Right panel and **b**), Mann–Whitney U test (**a** Left panel and **d**) or Kruskal–Wallis test with Dunn's test (**c**).

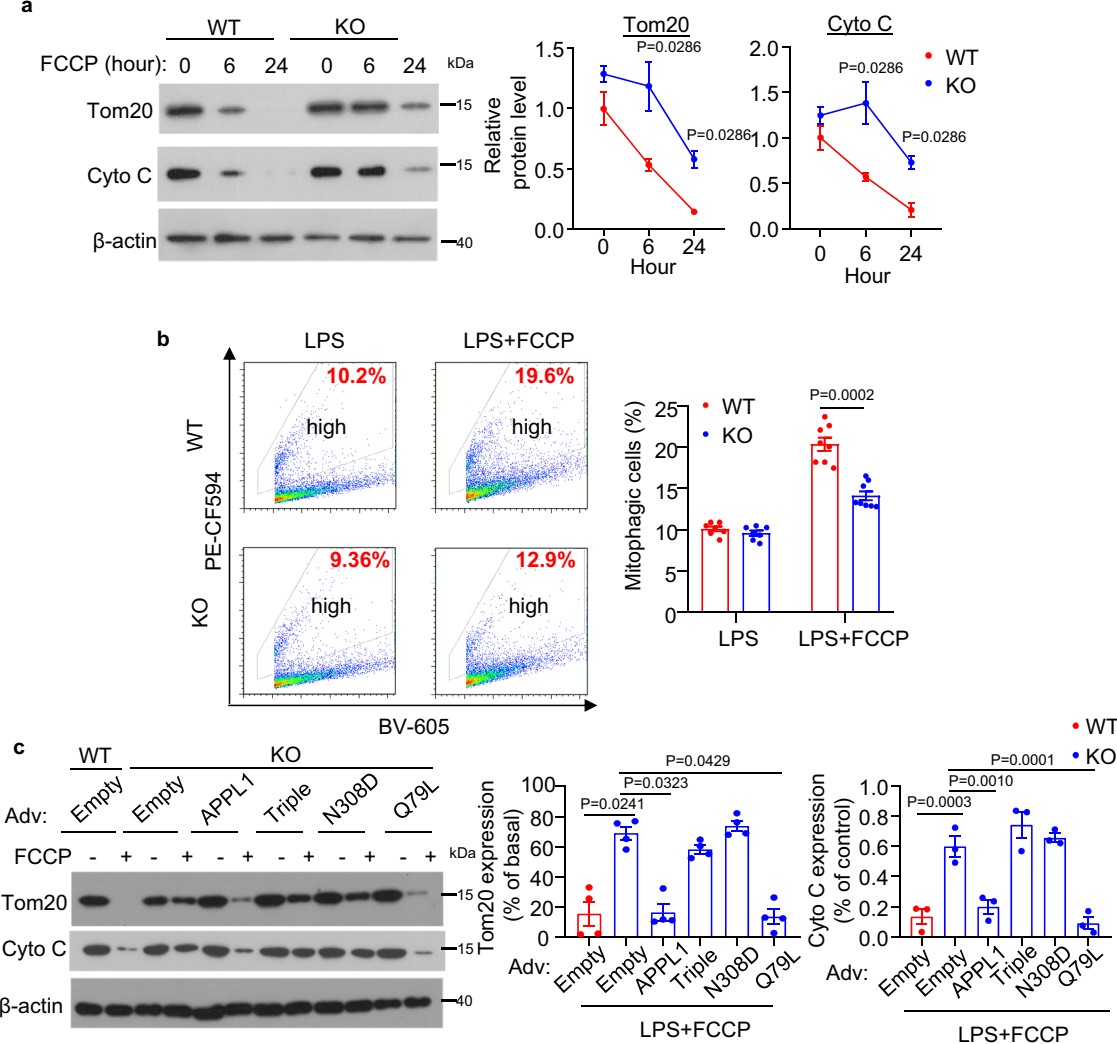

**Fig. 6 APPL1 deficiency blocks FCCP-induced mitophagy in BMDM. a–c** BMDM from APPL1 KO and WT littermates were used. **a** The BMDM with LPS priming and FCCP treatment for different timepoints were subjected to immunoblotting analysis of Tom20, cytochrome (Cyto C) and β-actin. The line charts are densitometric analysis of Tom20 and Cyto C normalized with β-actin. $n = 4$ biologically independent samples. **b** The BMDM were infected with adenovirus encoding mitochondrial target-Keima (Mito-Keima) for 48 h. The cells were then primed with LPS for 20 h, followed by stimulation with FCCP for 6 h. Flow cytometry analysis of mitophagic cells using Mito-Keima as described in Supplementary Fig. 4. LPS: $n = 7$, LPS + FCCP: $n = 8$ biologically independent samples. **c** The BMDM were infected with recombinant adenovirus encoding wild-type APPL1, APPL1 triple mutant (Triple), APPL1 N308D mutant or empty adenovirus as control for 24 h, followed by priming with LPS for 20 h. The LPS-primed BMDM were stimulated with ATP for 4 h or treated with PBS as control. Immunoblotting analysis of Tom20, cytochrome C (Cyto C), and β-actin level in total cell lysate. $n = 4$ biologically independent samples. The samples of immunoblotting analysis were collected from the same experiment and blots were processed in parallel. Data are displayed as mean ± SEM. Statistical significance was tested using Mann–Whitney U test (**a**, **b**), one-way ANOVA with post-hoc Bonferroni correction (**c** right panel) or Kruskal–Wallis test with Dunn's test (**c** left panel).

mitophagosomes, we measured colocalization of LC3+ puncta with MitoTracker in the presence of chloroquine (an autophagy inhibitor that blocks autophagosome fusion with lysosomes[37]). As expected, treatment with chloroquine further increased LC3+ puncta and MitoTracker colocalization in WT BMDM with NLRP3 inflammasome activation, but the increment in LC3-mitochondria colocalization was significantly reduced in APPL1 KO BMDM (Fig. 5c), indicating the defective mitophagic flux. Consistently, a marked reduction of colocalization of LAMP1 with MitoTracker was found in APPL1 KO BMDM in the presence or absence of chloroquine (Fig. 5d). NLRP3 agonists (such as ATP) induce p62 (the autophagy receptor) recruitment to the mitochondria via Parkin-dependent manner in macrophages[11]. Consistent to the previous study[11], immunofluorescence staining and immunoblotting analysis showed that p62 was recruited and

clustered onto mitochondria in LPS-primed WT BMDM upon stimulation with ATP, but this did not occur in APPL1 KO BMDM (Supplementary Figs. 5a and 6a). Taken together, these findings suggest that APPL1 deficiency selectively impairs the trafficking of mitophagosomes to lysosome for degradation, leading to accumulation of damaged mitochondria.

A recent study demonstrated that recruitment of NLRP3 to the dispersed trans-Golgi network is indispensable for NLRP3 activation[38]. Consistent to the previous study[38], trans-Golgi (visualized by TGN38) was disrupted and the colocalization of TGN38 and NLRP3 was increased in WT BMDM upon LPS + ATP stimulation, whereas genetic deletion of APPL1 had no detectable effect on NLRP3 accumulation on trans-Golgi network (Supplementary Fig. 5b).

**APPL1+ endosomes are recruited to mitochondria upon NLRP3 inflammasome activation via the interaction with Rab5.** APPL1 is an early endosomal protein with intracellular trafficking properties[25,39–41]. We examined whether subcellular localization and protein expression of APPL1 changes upon NLRP3 inflammasome stimulation. Similar to the changes in Tom20 and Cyto C expression, APPL1 protein expression drastically decreased in BMDM upon exposure with LPS + ATP, suggesting that APPL1 might target for mitophagy (Fig. 7a). Subcellular fractionation and subsequent immunoblotting analysis revealed that APPL1 and Rab5 were recruited to the mitochondrial fraction, coincided with the recruitment of p62 in BMDM in response to LPS + ATP stimulation (Fig. 7b). Consistently, immunofluorescence staining visualized that APPL1 colocalized with Rab5+ endosomes, and their colocalization moved to the perinuclear region and mitochondria in BMDM upon NLRP3 agonist stimulation in a time-dependent manner (Fig. 7c and Supplementary Fig. 7a). The APPL1+ or Rab5+ endosomes could be detected in mitophagasomes upon NLRP3 inflammasome activation (Supplementary Figs. 7b and 8a, b). On the contrary, there was little colocalization between APPL1 or Rab5 with the mitochondrial autolysosome (labelled with LAMP1 and MitoTracker in Supplementary Fig. 7c and LAMP2A and MitoTracker in Supplementary Fig. 8h) in these cells, and hence quantification of these two vesicle populations were not performed in the subsequent experiments.

We tested whether the interaction between Rab5 and APPL1 is required for recruitment of APPL1+ endosome to the mitochondria. Mutation of asparagine residue of APPL1 at position 308 to aspartic acid (N308D) in the BAR domain disrupts the interaction between APPL1 and Rab5[39]. APPL1 KO BMDM and WT BMDM were infected with adenovirus encoding human full-length wild-type APPL1, APPL1 N308D mutant or empty control. Confocal images indicated that a portion of full-length wild-type APPL1 colocalized with Rab5 and effectively recruited to the mitochondria, whereas the APPL1 N308D mutant did not (Supplementary Fig. 6b, c). To further validate the above observation, we generated an APPL1 mutant without endosomal localization ability by mutating arginine residue at 147 and lysine residues at 153 and 155 in the BAR domain of APPL1 to alanine (so-called APPL1 triple mutant)[27]. Like the N308D mutant, APPL1 triple mutant failed to recruit to Rab5+ endosomes and the mitochondria in BMDM under basal or NLRP3 stimulated conditions (Supplementary Fig. 6b, c). Consistently, subcellular fractionation experiment confirmed that the wild-type APPL1 were recruited to the mitochondrial fraction upon NLRP3 inflammasome activation, and this response was diminished in the APPL1 mutants (N308D and triple) (Supplementary Fig. 6a). Therefore, mitochondrial recruitment of APPL1 requires its endosomal localization and Rab5 binding upon NLRP3 inflammasome activation.

**Endosomal APPL1 and its interaction with Rab5 are important for the regulatory effects of APPL1 on mitophagy and NLRP3 inflammasome activity.** Under stress condition, Rab5+ endosomes are recruited to damaged mitochondria where they mediate Parkin-dependent mitophagy[23,42]. We speculated that APPL1 controls mitophagy and inflammasome activation via Rab5. Upon NLRP3 activation, recruitment of Rab5+ endosomes to the mitochondria (MitoTracker+), mitophagosomes (Mito-Tracker+LC3+) as well as formation of Rab5+ puncta were increased in WT BMDM, whereas genetic ablation of APPL1 altered these Rab5 trafficking, indicating that APPL1 regulates endosomal-dependent mitophagy (Fig. 7d and Supplementary Fig. 8a & e, f). Total number of mitophagosomes (LC3+MitoTracker+) and mitochondrial autolysosome were increased and decreased, respectively, in APPL1 KO BMDM (indicates defective mitophagy flux and fusion of mitophagosome with lysosome-Supplementary Fig. 8f–i). The increased number of mitophagosomes in APPL1 KO BMDM was mainly from Rab5- mitophagosomes, whereas the number of Rab5+ mitophagosomes was comparable between APPL1 KO and WT BMDM under LPS + ATP condition (Supplementary Fig. 8a-c). However, when the data are presented as percentage of total mitophagosome population, we found that the percentage of Rab5+ mitophagosomes was indeed lower in APPL1 KO BMDM than that in WT BMDM under LPS + ATP condition (Supplementary Fig. 8d).

Next we investigated whether the interaction between Rab5 and APPL1 is essential for the regulatory action of APPL1 on NLRP3 inflammasome and mitophagy. APPL1 KO BMDM and WT BMDM were infected with adenovirus encoding wild-type APPL1, APPL1 N308D mutant, APPL1 triple mutant or empty control, followed by assessment of inflammasome and mitophagy activities. APPL1-deficient BMDM with replenishment of wild-type APPL1 displayed a significant reduction of IL-1β secretion and caspase-1 activity after ATP or FCCP stimulation, when compared to those infected with empty adenovirus (Fig. 8a–d). The changes were accompanied by the restoration of mitophagy in APPL1 KO BMDM infected with adenovirus expressing wild-type APPL1, which was assessed by (1) immunoblotting of the mitochondrial protein (Tom20 and Cyto C) degradation (Figs. 9a and 6c) and (2) Mito-Keima mitophagy assay (Fig. 9c) as well as (3) p62 recruitment to the mitochondria (Supplementary Fig. 6a). Similarly, under ATP stimulation, recruitment of Rab5+ endosomes to the mitochondria in APPL1 *null* macrophages was restored by adenovirus expressing wild-type APPL1 (Fig. 9b and Supplementary Fig. 6a). On the contrary, the rescue effects of APPL1 replenishment on early endosome-mediated mitophagy and suppression of NLRP3 inflammasome were abrogated by APPL1 N308D mutant or APPL1 triple mutant (Figs. 6c, 8, 9 and Supplementary Fig. 6a). Next we determined whether activation of Rab5 is able to rescue the APPL1 *null* phenotypes. Adenoviral-mediated expression of constitutively active form Rab5 (Q79L) restored mitophagy and prevented the excessive caspase-1 activation and IL-1β secretion in APPL1-deficient BMDM (Figs. 6c, 8, 9 and Supplementary Fig. 6a). Collectively, these data indicate that both endosomal localization of APPL1 and its interaction with Rab5 are essential for the regulatory effects of APPL1 on mitophagy and inflammasome.

**APPL1 prevents inflammasome activation in mouse models with obesity and sepsis.** To validate the physiological relevance of our in vitro findings, we employed two different animal models with NLRP3 inflammasome activation in macrophages. The first is high-fat diet (HFD)-induced chronic and systemic inflammation model[43]. APPL1 is documented as a key regulator of insulin and adiponectin signalling in multiple tissues[44–46]. To exclude the metabolic effect of APPL1 in non-immune cells, we generated a hematopoietic-specific APPL1 KO mouse model by transplanting bone marrow cells isolated from APPL1 KO mice and WT controls into 6-week-old lethally irradiated male C57BL/6 N mice (so-called BMT-APPL1 KO and BMT-WT mice, respectively). PCR analysis of DNA extracted from peripheral blood cells confirmed successful reconstitution of bone-marrow-derived blood cells after the transplantation (Supplementary Fig. 9a). After feeding with a HFD for 36 weeks, both BMT-APPL1 KO mice and BMT-WT controls developed obesity and hyperlipidemia to a similar extent, when compared with their corresponding standard chow (STC)-fed controls (Supplementary Fig. 9b, c and

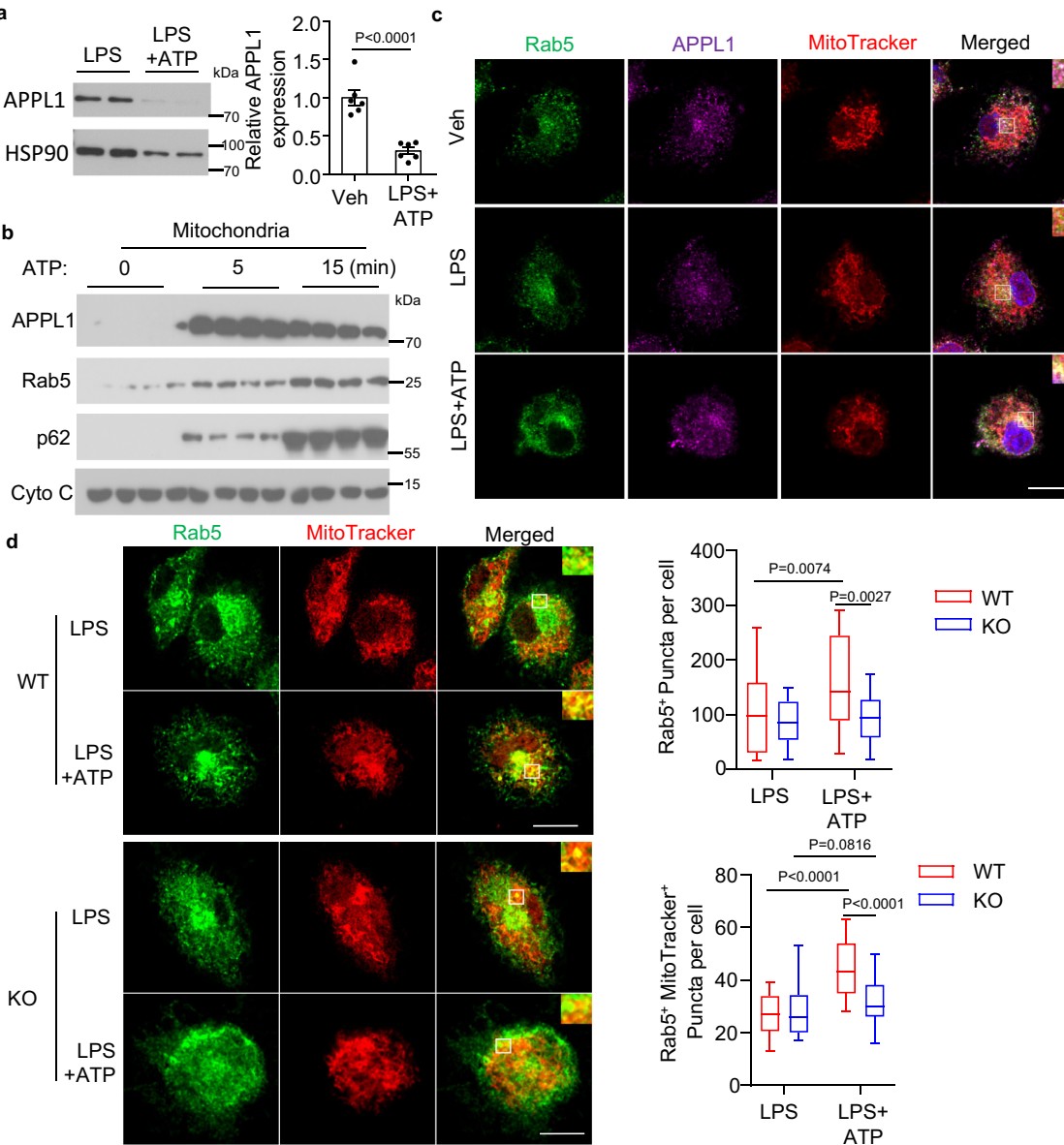

**Fig. 7 APPL1 promotes Rab5 recruitment onto mitochondria upon NLRP3 agonist stimulation. a–c** BMDM from C57BL/6 J mice were used. **a** The LPS-primed BMDM were stimulated with or without ATP for 4 h. Detection of APPL1 and HSP90 in the cell lysates by immunoblotting. The bar chart is densitometry analysis of APPL1 normalized with HSP90. $n = 6$ biologically independent samples. Data are displayed as mean ± SEM. The samples were collected from the same experiment and blots were processed in parallel. **b** The LPS-primed BMDM were treated with ATP as indicated, followed by mitochondrial fractionation and immunoblotting analysis. **c** The BMDM with or without LPS priming were stimulated with ATP for 1 h. BMDM without any stimulation is labelled as "Veh". MitoTracker Deep Red (Red) was added 30 min prior to collection of the BMDM, followed by fixation and immunofluorescence staining of Rab5 (green) and APPL1 (purple). Scale bar: 10 µm. **d** LPS-primed BMDM from APPL1 KO mice and their WT littermates were stimulated with ATP for 1 h. Immunofluorescence staining of Rab5 (green) and visualization of mitochondria by MitoTracker were performed as above. The boxplots are quantification of Rab5⁺ puncta (upper panel) and Rab5 and MitoTracker double positive puncta (lower panels). $n = 50$ biologically independent cells. Scale bar: 10 µm. Representative images were shown. In box and whisker plots, the whiskers extend to the minimum and maximum values, while the box presents the 25th and 75th percentiles with the central line at the median. Statistical significance was tested using two-tailed student's *t*-test (**a**) or Kruskal–Wallis test with Dunn's test (**d**).

Supplementary Table 1). Systemic inflammation, as reflected by circulating level of IL-1β and MCP-1, was increased by hematopoietic deletion of APPL1 (Fig. 10b and Supplementary Table 1). The anti-inflammatory adipokine adiponectin was modestly decreased in BMT-APPL1 KO mice when compared to BMT-WT controls under HFD feeding, although it did not reach statistical significance (Supplementary Table 1). H&E staining revealed that a massive infiltration of immune cells in epididymal white adipose tissue (eWAT) in BMT-APPL1 KO mice when compared to

their BMT-WT controls under HFD feeding (Fig. 10a). Noticeably, adipose tissue inflammation is a well-established mediator for obesity-induced metabolic disorders[47]. The expressions of inflammasome markers IL-1β and cleaved caspase-1 protein were higher in eWAT and the liver of BMT-APPL1 KO mice than that in BMT-WT mice under HFD feeding (Fig. 10c, e). Immunofluorescence staining revealed a notable enhancement in F4/80⁺ macrophage infiltration and caspase-1 expression in F4/80⁺ cells in eWAT of BMT-APPL1 KO mice than those in BMT-WT

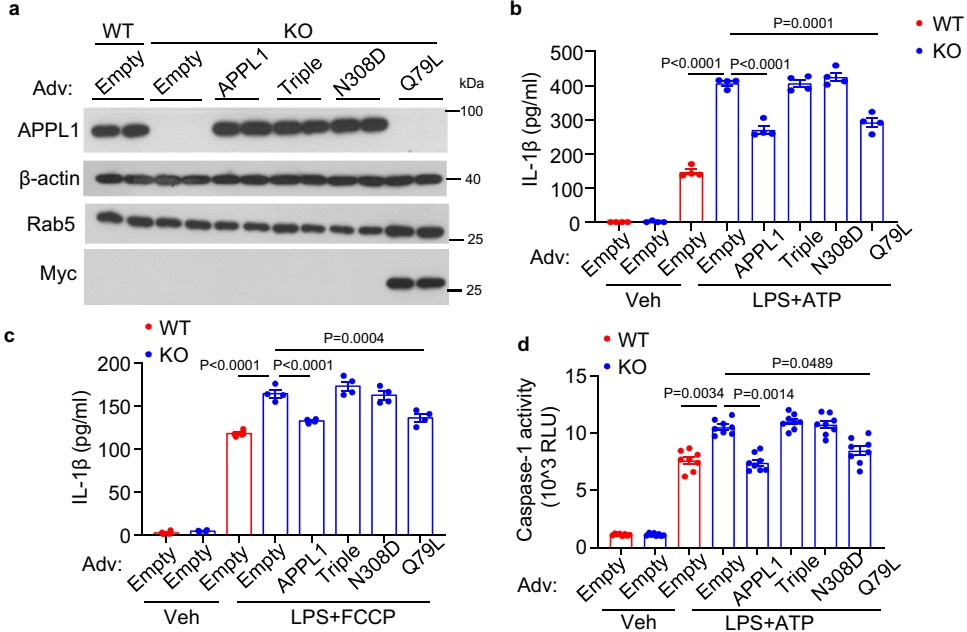

**Fig. 8 Endosomal and Rab5 binding ability of APPL1 are essential for its inhibitory action on NLRP3 inflammasome activation.** BMDM were isolated from 10–12-week-old APPL1 KO mice and their WT controls, followed by infection with recombinant adenovirus (Adv) expressing wild-type APPL1, APPL1 triple mutant (Triple; without endosomal localization ability), and APPL1 N308D mutant (without Rab5 binding ability), constitutive active form of Rab5 (Q79L) with Myc epitope or empty vector as control for 48 h. **a** Immunoblotting analysis of APPL1, Rab5, β-actin, and Myc in the infected BMDM. The samples of immunoblotting analysis were collected from the same experiment and blots were processed in parallel. **b**–**d** The infected BMDM were primed with LPS for 20 h, followed by stimulation with ATP for 4 h (**b**, **d**) or FCCP for 6 h (**c**). IL-1β in cell culture supernatant after ATP (**b**) or FCCP (**c**) treatment. $n = 4$ biologically independent samples. **d** Caspase-1 activity in cell culture supernatant after ATP stimulation. $n = 8$ biologically independent samples. Data are displayed as mean ± SEM. Statistical significance was tested using one-way ANOVA with post-hoc Bonferroni correction (**c**) or Kruskal–Wallis test with Dunn's test (**b**, **d**).

controls under HFD feeding (Fig. 10f). Hematopoietic-specific deletion of APPL1 aggravated HFD-induced glucose intolerance and insulin resistance, as assessed by glucose tolerance test (GTT) and insulin tolerance test (ITT), respectively (Supplementary Fig. 10). In vitro, we tested whether APPL1 deficiency has any effect on inflammasome activation induced by palmitic acid, the most abundant saturated fatty acid with marked elevation in obesity and is known to induce NLRP3 inflammasome activation by inhibiting autophagy/mitophagy[48,49]. As expected, palmitic acid markedly induced caspase-1 activation and IL-1β production in WT BMDM primed with LPS, whereas genetic deletion of APPL1 further potentiated these inflammasome responses (Supplementary Fig. 11a, c). APPL1 deficiency had no effect on palmitic acid-induced TNF-α secretion (Supplementary Fig. 11b). Consistent to the findings using other NLRP3 inflammasome activators, APPL1 deficiency further enhanced palmitic acid-induced loss of mitochondrial membrane potential (Supplementary Fig. 11d). Collectively, the above findings showed that APPL1 deficiency aggravates inflammasome activation in response to lipotoxicity, which exacerbates obesity-mediated insulin resistance.

To further confirm the in vivo relevance of control of NLRP3 inflammasome by APPL1, we injected LPS into BMT-APPL1 KO mice and BMT-WT mice to induce septic shock. In line with our in vitro findings, the protein levels of IL-1β and IL-18 but not MCP-1 in serum and peritoneal fluid were significantly higher in BMT-APPL1 KO mice than in BMT-WT mice after injection with LPS for 6 h (Supplementary Fig. 12a–e). The detrimental effects of LPS were more pronounced in BMT-APPL1 KO mice when compared to BMT-WT mice, as exemplified by more obvious drop in body weight and core temperature (Supplementary Fig. 12f–g).

## Discussion

Mitophagy maintains quality and proper function of mitochondria. Defective mitophagy causes persistent and unresolved NLRP3 inflammasome activation, which in turn leads to both autoinflammatory and metabolic diseases[6,11–13]. In this study, we uncovered that APPL1 mediates the interconnectivity between endosomal pathway and mitophagy, which is required to restrict NLRP3 inflammasome in macrophages (Supplementary Fig. 13). APPL1 deficiency in hematopoietic cells exacerbates obesity-induced chronic inflammation and endotoxin-induced septic shock in animal models.

Emerging evidence indicate that the early endosomes regulate autophagy/mitophagy under stress condition[50]. Inactivation of early endosome function by downregulation of COPI subunits inhibits autophagy and leads to accumulation of autophagosomes and p62[51]. Rab5+ early endosome captures damaged mitochondria labelled with ubiquitinated proteins for lysosomal degradation and clearance via a Parkin-dependent pathway[23,42]. Overexpression of dominant negative form of Rab5$^{S34N}$ (a GDP-bound mutant) abolished FCCP-induced mitophagy in mouse embryonic fibroblasts, leading to excessive cell death[23]. Although these studies showed the key role of early endosomes in mitochondrial homeostasis, to the best of our knowledge, whether and how the interplay between the early endosomes and mitophagy controls inflammasome activation are unknown. We showed that NLRP3 inflammasome-induced recruitment of Rab5+ early endosomes to the damaged mitochondria is attenuated by APPL1 deficiency, and this defect links to impaired formation of mitophagosomes and mitochondrial autolysosomes. The defective early endosome-mediated mitophagy causes excessive accumulation of damaged mitochondria, leading to generation of mtROS and cytosolic oxidized mtDNA, which act as second signals to

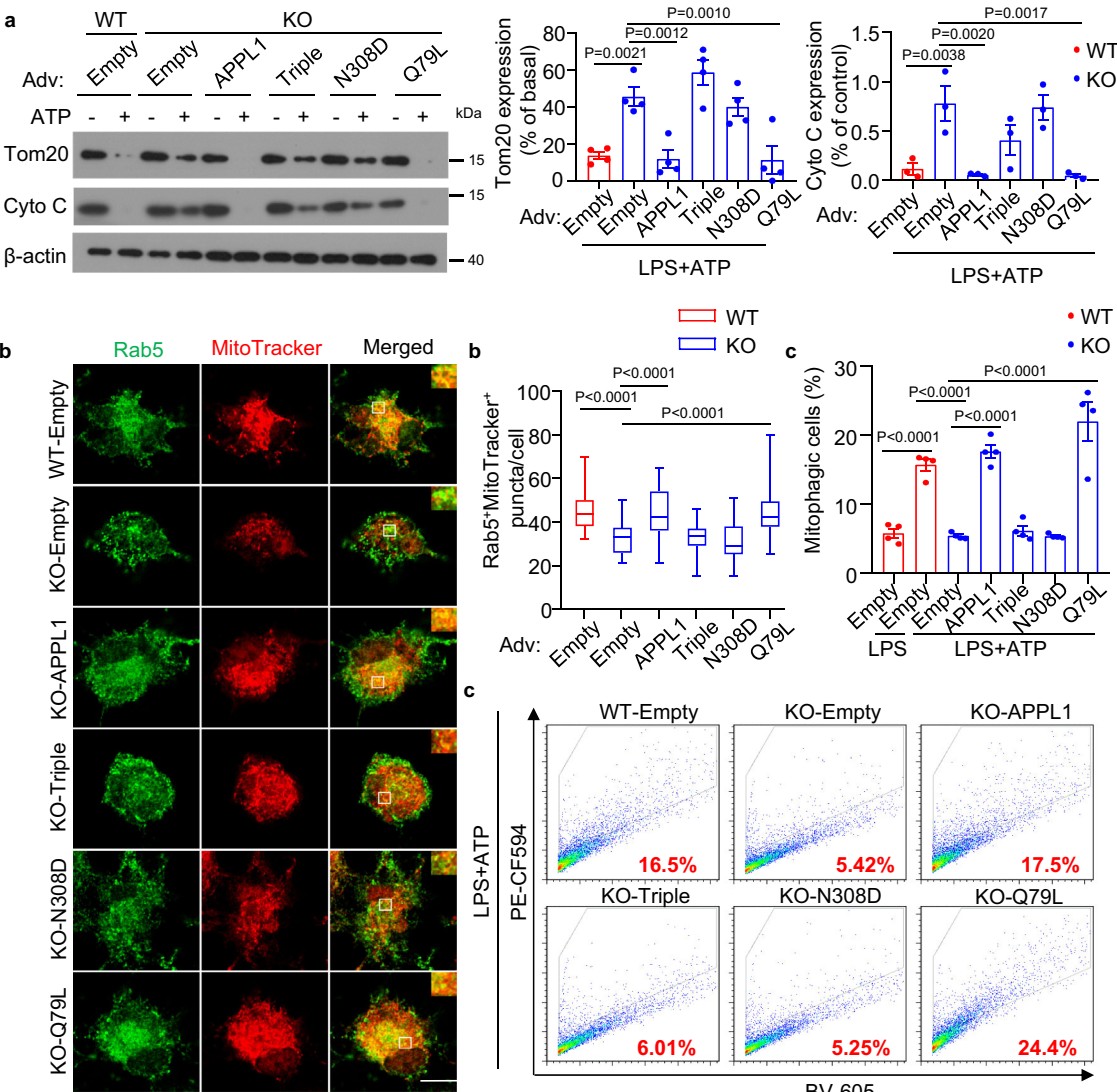

**Fig. 9 The promoting effect of APPL1 on mitophagy and Rab5 recruitment to mitochondria depends on its endosomal localization and Rab5 interaction ability.** BMDM from APPL1 KO mice and their WT controls were infected with the recombinant adenovirus expressing APPL1 or its mutants or empty vector as indicated, followed by priming with LPS and treatment with ATP for 4 h. **a** Total cell lysate was subjected to immunoblotting analysis of Tom20, Cyto C, and β-actin. The bar charts are densitometry analysis of relative Tom20 and Cyto C levels normalized with β-actin. $n = 4$ biologically independent samples. The samples were collected from the same experiment and blots were processed in parallel. **b** Immunofluorescences staining of Rab5 (Green) and MitoTracker (Red) in the LPS-primed BMDM after ATP stimulation for 1 h. Scale bar: 10 μm. The boxplot is quantification of colocalization of Rab5 and MitoTracker positive puncta. $n = 50$ biologically independent cells. **c** The BMDM were co-infected with the recombinant adenovirus encoding the Mito-Keima and Empty vector or APPL1 or its mutants as indicated, followed by LPS + ATP treatment for 30 min. Mitophagic cell population was determined using flow cytometry analysis as described in Supplementary Fig. 4. $n = 4$ biologically independent samples. Representative images were shown. Data are displayed as mean ± SEM. In box and whisker plots, the whiskers extend to the minimum and maximum values, while the box presents the 25th and 75th percentiles with the central line at the median. Statistical significance was tested using one-way ANOVA with post-hoc Bonferroni correction (**a**, **c**) or Kruskal–Wallis test with Dunn's test (**b**).

induce NLRP3 inflammasome hyperactivation. The modulating effects of APPL1 on inflammasome and mitophagy depend on its endosomal localization and binding ability to Rab5. Rab5 interacts with class C VPS/HOPS complex (the Rab7 guanine nucleotide exchange factor [GEF]) to activate Rab7, leading to conversion of early endosomes to late endosomes[52]. Rab7 promotes mitophagy by inducing translocation of ATG9a (the autophagy-related transmembrane protein) and mitophagosome formation as well as supporting recruitment of the DENN domain-containing heterodimer FLCN-FNIP1 to the damaged mitochondria via Parkin-dependent manner[24,53]. The regulatory action of Rab5 on Rab7 activation might also contribute to the control of mitophagy. Our data indicate that APPL1 deficiency abrogates p62 and Rab5 recruitment to the mitochondria and abolishes the trafficking mitophagosome to lysosome, all of which possibly contribute to the defective mitophagy and accumulation of damaged mitochondria, yet the precise step by which APPL1 controls mitophagy remains elusive. Further study investigating the detailed mechanism by which APPL1 controls the interplay between endosomal system, mitophagic machinery and inflammasome is warrant.

NLRP3 inflammasome can be activated by diverse stimuli[31]. The classical NLRP3 stimuli, such as ATP, nigericin, MSU, FCCP, rotenone, antimycin, and palmitic acid, are known to induce

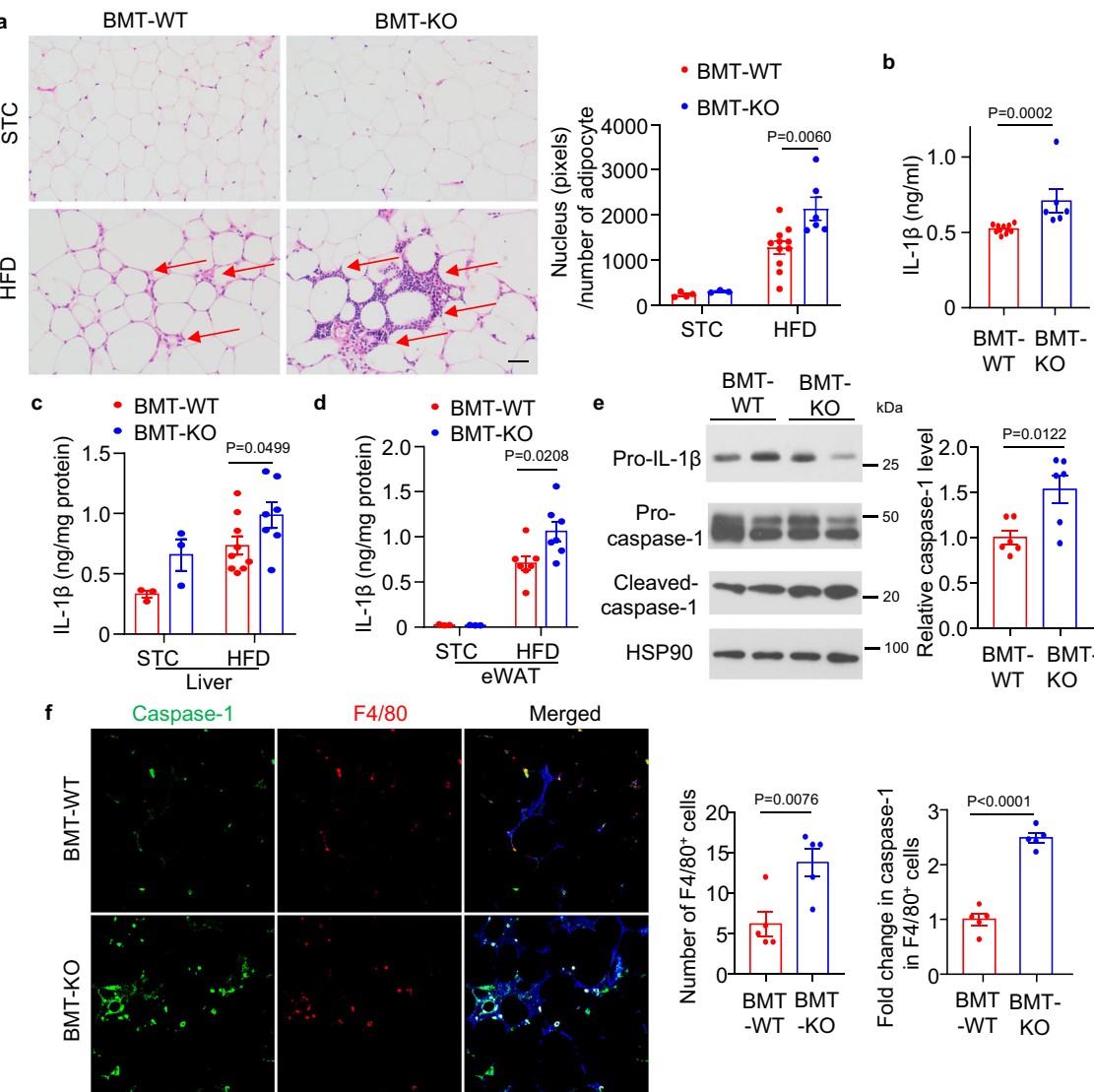

**Fig. 10 Ablation of APPL1 in hematopoietic cells exacerbates dietary-induced systemic inflammation and inflammasome activation in adipose tissue and liver.** Liver and epididymal white adipose tissue (eWAT) were collected from BMT-KO mice and BMT-WT controls fed with STC or HFD for 36 weeks. **a** H&E staining of eWAT. Scale bar: 100 μm. Arrows indicate immune cell infiltration. Quantification of number of nucleus per number of adipocyte to reflect immune cell infiltration. BMT-WT-STC: $n = 4$; BMT-KO-STC: $n = 3$; BMT-WT-HFD: $n = 11$; BMT-KO-HFD: $n = 6$ biologically independent animals. IL-1β levels in **b** serum (BMT-WT: $n = 11$; BMT-KO: $n = 6$ biologically independent animals), **c** liver (BMT-WT-STC: $n = 3$; BMT-KO-STC: $n = 3$; BMT-WT-HFD: $n = 9$; BMT-KO-HFD: $n = 7$ biologically independent animals.) and (**d**) eWAT measured by immunoassay. (BMT-WT-STC: $n = 3$; BMT-KO-STC: $n = 3$; BMT-WT-HFD: $n = 7$; BMT-KO-HFD: $n = 7$ biologically independent animals.). **e** Immunoblotting detection of caspase-1 and IL-1β in the liver. $n = 6$ biologically independent animals. $n = 6$ biologically independent animals. **f** Immunofluorescence staining of F4/80 (red) and caspase-1 (green) in eWAT. The nucleus was stained with DAPI (blue). Scale bar: 50 μm. Quantification of number of F4/80$^+$ cells and fold change in caspase-1 intensity in F4/80$^+$ cells are displayed in the right panels. Representative images are shown. $n = 5$ biologically independent animals. Data are displayed as mean ± SEM. Statistical significance was tested using two-tailed student's $t$-test (**a**, **c**, **d**, **e**, **f**) or Mann–Whitney U test (**b**).

mitochondrial dysfunction or generate mitochondrial danger signals in the immune cells[6,12,13,48]. In current study, we demonstrated that APPL1 deficiency provokes the mitochondrial damaging stimuli-induced IL-1β production in BMDM. On the contrary, APPL1 deficiency has no obvious effect on IL-1β secretion upon treatment with flagellin, Poly(dA:dT) or LPS transfection, arguing a specific role of APPL1 in restricting mitochondrial damaging stimuli-induced NLRP3 inflammasome activation. However, it should be noted that LPS transfection has also been shown to activate NLRP3 inflammasome via caspase-11-potassium efflux axis in macrophages[54]. Unfortunately, whether LPS transfection leads to increase of mitochondrial damages and/or danger signals in macrophage is currently unclear[54],

despite a recent study showing LPS transfection leads to an increase amount of cytosolic mtDNA by induction of the pore-forming protein Gasdermin D in endothelial cells[55]. We believe that defective mitophagy induced by aberrant APPL1-Rab5 axis leads to excessive accumulation of mitochondrial danger signals that augment NLRP3 activation in macrophages. Since intraperitoneal injection of LPS is also known to induce noncanonical inflammasome via caspase-11 in the immune cells of mice[56,57], the exact mechanism whereby APPL1 controls sepsis and IL-1β production in animal model requires further investigation.

Our study shows that the reciprocal regulation between APPL1 and Rab5 is essential to fine-tune endosome-mediated mitophagy and NLRP3 inflammasome activity. The facts that (1) endosomal

localization and Rab5 binding ability of APPL1 are required for Rab5 recruitment to the damaged mitochondria for lysosomal degradation, and (2) activation of Rab5 rescues the impairment of mitophagy and elevated inflammasome activation in APPL1 *null* macrophages, support a notion that APPL1 acts as an upstream regulator of Rab5. Consistent to our findings, a recent study also showed that knockdown of APPL1 expression reduces Rab5 activity in the cells with overexpression of β-Amyloid precursor protein[58], a protein known to trigger Alzheimer's disease. On the other hand, APPL1 also acts as a downstream effector of Rab5 because blocking the Rab5 binding ability of APPL1 abolishes the recruitment of APPL1 to mitochondria and the inhibitory effect of APPL1 on inflammasome.

Our findings indicate that APPL1[+] and Rab5[+] endosomes somehow sense damaged mitochondria. Majority of APPL1 co-localizes with Rab5[+] endosomes and evenly distributes inside the cells under basal condition, but recruits to the perinuclear region and mitochondria upon NLRP3 agonist stimulation. A previous study also demonstrated that APPL1 is recruited to perinuclear region and aggresomes under proteasomal stress via ubiquitination[59]. Consistent to our finding, two independent studies demonstrated that Rab5[+] endosomes are recruited to mitochondria upon activation of mitophagy by either treatment with valinomycin (the potassium ionphore) or the mitochondrial uncoupler[20,23]. The recruitment of Rab5 to mitochondria is believed to be controlled by its upstream regulator of Rab5-GEF and/or beclin-1[20,23]. On the other hand, in response to oxidative stress, APPL1 is recruited to nucleus but not mitochondria, whereas Rab5 together with its activator Alsin are recruited to mitochondria where they block cytochrome c release and prevent apoptosis[25,42]. Nevertheless, the mechanism(s) by which APPL1[+] and Rab5[+] endosomes sense mitochondrial damage and target to mitochondria in macrophage is currently unknown, but it is worth to explore the involvement of autophagy-related proteins such as beclin-1 and Rab5-GEF in future study.

Activity of NLRP3 inflammasome in adipose tissue positively correlates with obesity and its related type 2 diabetes in human[60]. In animal models, inactivation of NLRP3 prevents ageing- and obesity-induced chronic inflammation, glucose intolerance and nonalcoholic fatty liver disease[5,43,61,62]. Under obese condition, saturated fatty acids such as palmitic acid and its downstream metabolite ceramide induce inflammasome activation by inhibiting AMPK-mediated autophagy in macrophages or inhibiting PINK1-Parkin-mediated mitophagy in endothelial cells[43,48,49]. IL-1β derived from NLRP3 inflammasome impairs insulin sensitivity by eliciting phosphorylation of insulin receptor substrate at serine 307, a modification that blocks PI3K-Akt activation[48]. APPL1 mediates insulin and adiponectin actions in multiple metabolic tissues[63]. Our previous studies demonstrated that APPL1 deficiency causes insulin resistance in the liver and endothelium by inhibiting Akt activation[46,64]. The loss-of-function mutations of APPL1 are identified in familial diabetes[65]. In current study, we found that obesity-induced glucose intolerance is more severe in BMT-APPL1 KO mice than BMT-WT controls under HFD feeding. The glucose dysregulation in BMT-APPL1 KO mice is associated with increased levels of IL-1β in the circulation, visceral adipose tissues and the liver. However, whether these defects are solely due to APPL1 deficiency in macrophages or other immune cells and elevation of IL-1β, and occurrence of defective mitophagy in BMT-APPL1 KO remain elusive. In vitro, we showed that palmitic acid-induced caspase-1 activation, IL-1β production and mitochondrial dysfunction are also more pronounced in APPL1-deficient macrophages. The changes in NLRP3 inflammasome activity are believed as a major contributor to glucose intolerance in BMT-APPL1 KO mice. Consistent to our data, deletion of the

mitophagy receptor FUNDC1 exacerbates obesity-induced adipose tissue inflammation and glucose dysregulation[66]. Similar to those mice with defective mitophagic response[6,11–13,15,66], BMT-APPL1 KO mice are more sensitive to septic response and IL-1β production upon LPS injection. Collectively, these findings reveal that APPL1 in immune cells is crucial for preventing obesity- and LPS-induced inflammation, perhaps, by promoting mitophagy.

To conclude, our work indicates that APPL1 is essential for early endosome-mediated mitophagy, which suppresses NLRP3 inflammasome activation in macrophages. The actions of APPL1 in immune cells prevent obesity-associated metabolic disorders and endotoxin-induced sepsis. Future studies should (1) identify a mean of manipulation such as endosomal regulatory pathway for the# treatment of metabolic and autoimmune diseases and (2) validate whether this highly regulatory and dynamic endosomal and mitochondrial network is also altered in human immuno-metabolic diseases.

## Methods

**Animal studies.** Generation and maintenance of APPL1 KO mice and their WT littermates were described in our previous study[64]. Groupings of animals were according to their genotypes and thus no randomization was used. The researchers were not blinded to the grouping of the experimental animals. Animals were housed in a room with temperature control (22 ± 1 °C) and humidity control (60 ± 10%), along with a 12 h light-dark cycle in the animal facilities of The Hong Kong Polytechnic University (PolyU) and The University of Hong Kong (HKU). The mice were allowed to have a free access to water and food unless otherwise specified. To generate hematopoietic cells-specific APPL1 KO mice and WT controls (BMT-APPL1 KO and BMT-WT, respectively), we adopted bone marrow transplantation as previously described[67]. Briefly, bone marrow cells isolated from tibias and femurs of 10-week-old male APPL1 KO mice and their WT littermates were resuspended in phosphate buffered saline (PBS) with 2% fetal bovine serum (FBS, Catalog#10270106, Gibco). Recipient C57BL/6 N mice pretreated with acidified water (pH 3.0) were irradiated at the dosage of 11 Gy, followed by intravenous injection with $1 \times 10^7$ bone marrow cells one day after the irradiation. The mice were then allowed to recover for 6 weeks before feeding with a HFD (20 kcal% protein, 45 kcal% fat, and 35 kcal% carbohydrates; Catalog# D12492, Research Diets, Inc) or LPS injection. Genotyping was performed before dietary intervention to ensure successful engraftment. In brief, 100 μl of peripheral blood was collected from saphenous vein and mixed with 10 μl of 1% ethylenediamine-tetraacetic acid (EDTA) solution. The blood was then centrifuged at $1200 \times g$ for 10 min at 4 °C and the supernatant was discarded. The cell pellet was digested with DirectPCR Lysis Reagent (Catalog# 102-T, Viagen Biotech) supplemented with 0.5 mg/ml proteinase K (Catalog# P6556, Sigma–Aldrich) overnight at 55 °C, followed by incubation at 85 °C for 1 h before PCR with 2xEs Taq MasterMix (Dye) (Catalog# CW0690L, CWbiotech). Primers sequences for genotyping are listed in Supplementary Table 2. For GTT, mice were fasted for 16 h, followed by intra-peritoneal injection of D-glucose solution. For ITT, mice were fasted for 6 h and intraperitoneally injected with recombinant insulin (Catalog# 91077 C, Sigma–Aldrich). The dosage of glucose and insulin used were indicated in the figure legends. Blood glucose was measured using a glucometer (ACCU-Check Performa, Roche). For septic shock model, 10 mg/kg of LPS from *Escherichia coli* O111:B4 (Catalog# LPS25, Sigma–Aldrich) was intraperitoneally injected into BMT-APPL1 KO mice and BMT-WT controls. Core body temperature was measured using a thermometer with a rectal probe (Model 4610 Precision Thermometer, Measurement Specialties). All animal experimental protocols were approved by Animal Subjects Ethics Sub-Committee at PolyU and the Committee on the Use of Live Animals in Teaching and Research at HKU.

**Isolation, culture, and treatments of BMDM.** L929 conditioned medium was prepared by culturing L929 mouse fibroblasts (ATCC® CCL1™) in DMEM (Catalog# 12800082, Gibco) supplemented with 10% FBS and 1% penicillin (100 U/ml) and streptomycin (100 μg/ml) (PS, Catalog# 15140163, Gibco) at confluence for 3 days. Bone marrow cells from tibias and femurs of APPL1 KO mice and WT littermates were cultured with DMEM supplemented with 10% FBS, 20% L929 conditioned medium, and 1% PS for 7 days. The differentiated BMDM were then detached using cell dissociation buffer (Catalog# 13151014, Gibco) and cultured in DMEM supplemented with 10% FBS and 1% PS. To induce NLRP3 inflammasome activation, BMDM were first primed with LPS (100 ng/ml) for 20 h, followed by stimulation with ATP (3 mM) for 4 h. To trigger NLRC4 inflammasome activation, LPS-primed BMDM were transfected with 20 μg/ml flagellin (Catalog# tlrl-epstfla, Invivogen) using Lipofectamine® 3000 reagent (Catalog # L3000, Invitrogen) and incubated for 8 h[68,69]. To induce inflammasome via AIM2, Poly(dA:dT) was delivered into LPS-primed BMDM using Lipofectamine® 3000 reagent at the dose of 2 μg/ml and incubated for 6 h[8]. To provoke noncanonical NLRP3 inflammasome activation, BMDM were transfected with 2 μg/ml LPS using Lipofectamine® 3000

reagent and incubated for 6 h[70]. To induce mitochondrial damage and subsequent inflammasome activation, LPS-primed BMDM were treated with 25 μM FCCP (Catalog# 15218, Cayman Chemical), 40 μg/ml antimycin A (Catalog# 19433, Cayman Chemical) or 10 μM rotenone (Catalog# 13995, Cayman Chemical) for 6 h. To induce M1 macrophages polarization, BMDM were treated with 100 ng/ml LPS and 100 ng/ml interferon gamma (IFNγ) (Catalog# 485-MI, R&D systems) for 24 h. For the palmitic acid treatment, palmitic acid (Catalog# 10006627, Cayman Chemical) was dissolved in 0.1 M NaOH at 65 °C to prepare a 100 mM stock solution. Palmitic acid solution was conjugated with 5% fatty acid free, low endotoxin bovine serum album (BSA) (Catalog# A8806, Sigma–Aldrich) to obtain a 2.5 mM working solution. BMDM were treated with 0.5 mM palmitic acid conjugated BSA for 24 h to stimulate NLRP3 inflammasome activation. To block endosomal acidification and mitophagy, LPS-primed BMDM were pretreated with 20 μM chloroquine diphosphate salt (CQ) (Catalog# C6628, Sigma–Aldrich) for 1 h before subsequent stimulation with inflammasome activator. To inhibit cGAMP synthase (cGAS), LPS-primed BMDM were pretreated with 1 μM of cGAS small molecule inhibitor RU.521 (Catalog# 31765, Cayman Chemical) for 4 h before the inflammasome activation. Two hundred micromolar of 8-OH-dG (Catalog# 89320, Cayman Chemical) was added to BMDM 3 h before ATP stimulation to inhibit NLRP3 inflammasome.

*Small interfering RNA (siRNA) transfection.* siRNAs (GenePharma) were mixed with DharmaFECT 3 transfection reagent (Catalog# T-2003-03, Dharmacon) per the manufacturer's instruction and added to the BMDM at a final concentration of 20 nM[71]. After incubation with the siRNA complex for 24 h, BMDM were primed with LPS for 20 h and then stimulated with inflammasome inducers as indicated in the figure legends. Details of the sequences of siRNA are listed in Supplementary Table 4.

**Generation and infection of adenoviruses.** Adenoviruses encoding wild-type APPL1 and APPL1 (R147A, K153A, and K155A) triple mutant were generated using site-directed mutagenesis kit (Catalog# 200523, Agilent) and purified using an adenovirus purification kit (Catalog# 240243, Agilent) as previously described[39]. Adenoviruses encoding APPL1 (N308D), Rab5(Q79L) mutants and mitochondria targeting Mito-Keima were generated and purified by Vigene Biosciences. The mitochondria targeting sequence from subunit VIII of human cytochrome C oxidase was added to the 5′ end of Mito-Keima-Red and the expression of Mito-Keima is driven by cytomegalovirus promoter[35]. BMDM were infected with the adenovirus at 100 multiplicity of infection (MOI) for 24 h, followed by priming and inflammasome activation as described above.

**Flow cytometry analysis of macrophage differentiation efficiency and mitophagy.** To examine macrophage differentiation efficiency, the bone marrow cells after incubation with L929 conditioned medium for 7 days were collected and stained with a PE-anti-F4/80 antibody (Catalog# 123110, BioLegend) and FITC-anti-CD11b antibody (Catalog# 101206, BioLegend) for 30 min on ice in the dark. The negative controls are the respective isotype controls (Catalog# 400508 and 400634, BioLegend). The cells were then washed and subjected to flow cytometry analysis on FACSAria™. The live cells were gated based on their forward scatter and side scatter, followed by the F4/80 and CD11b expression. For assessment of mitophagy, BMDM infected with adenovirus encodes Mito-Keima or the empty vector as control with or without NLRP3 inflammasome activation were subjected to flow cytometry analysis with FACSAria™ III equipped with 405-nm and 561 nm lasers. The live cells were gated based on their forward scatter and side scatter, followed by the selection of cells that express Mito-Keima based on their fluorescence in the BV605 channel compared to the BMDM infected with the control adenovirus. The detailed gating strategy and representative results were shown in Supplementary Fig. 4. Mitophagic cells with a high ratio of emission at PE-CF594/BV605 were selected and analyzed as previously described[36].

**Real-time quantitative PCR.** Total RNA was extracted with TRIzol reagent (Catalog# 15596026, Invitrogen). One μg RNA was reversely transcribed using GoScript™ Reverse Transcription Mix from Promega (Catalog# A2801). Real-time Quantitative PCR (QPCR) was performed with ViiA 7 Real-Time PCR System (Applied Biosystems) using QuantiNova SYBR® Green (Catalog# 208056, QIA-GEN) with the primers targeting different genes (Supplementary Table 2).

**Biochemical and immunological assays.** Level of IL-1β in tissue homogenate, circulation and cell culture supernatant were quantified with mouse IL-1 beta/IL-1F2 DuoSet ELISA (Catalog# DY401, R&D Systems). IL-18, TNF-α, and MCP-1 in cell culture supernatant, serum and peritoneal lavage fluid were measured using mouse IL-18 ELISA (Catalog# BMS618-3, Invitrogen), mouse TNF-α DuoSet ELISA (Catalog# DY410, R&D Systems) and mouse MCP-1 DuoSet ELISA (Catalog# DY479, R&D Systems), respectively. Insulin and adiponectin in mouse serum were quantified using ELISA kits from Antibody and Immunoassay Services, HKU (Catalog# 32270 and 32010). Caspase-1 activity was determined using Caspase-Glo® 1 Inflammasome Assay (Catalog# G9951, Promega). All the above assays were performed according to manufacturer's instructions.

**In vitro assays for mitochondrial function.** To measure mitochondrial mass in LPS-primed BMDM, total DNA was extracted using DNeasy Blood & Tissue Kit (Catalog# 69504, QIAGEN) according to the manufacturer's instruction. Mitochondrial DNA was quantified by QPCR using primers targeting the D-loop region of mtDNA, and its relative expression was normalized with nuclear DNA Tert. Protein expression of the mitochondrial proteins including Tom20, Cyto C, and pyruvate dehydrogenase were assessed by immunoblotting approach[14]. Mitochondrial membrane potential in BMDM was determined using TMRE membrane potential assay (Catalog# 701310, Cayman Chemical) with 50 nM TMRE. Fluorescence was measured using BMG LABTECH CLARIOstar with excitation and emission wavelengths at 530 and 580 nm, respectively. For ROS measurement, BMDM were incubated with CM-H2DCFDA (10 μM, catalog# C6827, Invitrogen) or MitoSOX (5 μM, catalog# M36008, Invitrogen) for 30 min after LPS priming, followed by ATP treatment for 30 min. Mitochondrial damage was assessed with co-staining using MitoTracker Deep Red (500 nM) and MitoTracker Green (200 nM) (Catalog# M7514 & M22426, Invitrogen) in the last 15 min of ATP treatment. Stained cells were subjected to flow cytometry analysis using BD FACSAria™ III.

**Immunoblotting.** Tissues or cells were homogenized in a RIPA lysis buffer (150 mM NaCl, 50 mM Tris HCl [pH 7.4], 2 mM EDTA, 0.1% SDS, 1% NP-40) supplemented with protease inhibitors (Catalog# HY-K0010, MedChemExpress) and phosphatase inhibitor cocktail (Catalog# B15001, Bimake). Proteins were separated using SDS-PAGE and transferred onto polyvinylidene difluoride membrane (Catalog# 1620177, BIO-RAD). The membrane was blocked with 10% nonfat milk for an hour, followed by incubation with primary antibody at 4 °C overnight. The membrane was washed with TBST (2.7 mM Tris base, 137 mM NaCl and 0.1% Tween 20) and then incubated with corresponding secondary antibody conjugated with horseradish peroxidase (Cell Signaling Technology) for 1 h at room temperature. Details of primary and secondary antibodies are listed in Supplementary Table 3. After washing with TBST, the proteins were visualized using enhanced chemiluminescence reagents (BIO-RAD) and quantified using ImageJ.

**Histological analysis and immunofluorescence staining.** Tissues were fixed in neutral buffered formalin overnight and processed with Excelsior™ AS Tissue Processor (Thermo Scientific) as we previously described[72]. The tissues were then cut into 5 μm section using microtome, followed by staining with hematoxylin and eosin (H&E) and analyzed with ImageJ. For immunofluorescence staining, antigen retrieval of the sections was done by boiling in sodium citric buffer (0.1 mol/L sodium citrate, 0.1% Tween 20, pH 6.0) for 20 min, followed by blocking with 5% FBS in PBS for 1 h at room temperature, overnight incubation with primary antibodies at 4 °C and fluorescent dye-conjugated anti-rabbit, anti-mouse, or anti-rat IgG secondary antibodies at room temperature for 1 h.

For immunofluorescence staining of BMDM, $2 \times 10^5$ cells were seeded onto sterile 20 mm coverslips and cultured in a 6-well plate, followed by the ATP treatment as follows (1) 10 min for Supplementary Fig. 5b; (2) 30 min for Supplementary Fig. 8a; (3) 60 min for Figs. 5c, d, 7c, d 9b, Supplementary Figs. 5a, 6b, c, 7c and 8h), and (4) 5, 10, and 15 min time course for Supplementary Fig. 7a and b . The treatment time is also described in each figure legend. BMDM were incubated with MitoTracker Deep Red (500 nM) in the last 30 min of the treatment and then fixed in 10% paraformaldehyde for 15 min or ice-cold methanol in −20 °C (for Supplementary Figs. 5b and 7b) for 15 min. The cells were then washed with PBS and blocked with 5% FBS and 0.03% Triton X-100 in PBS (or without Triton X-100 for methanol fixed cells) for 1 h. Cells were sequentially stained with primary antibodies overnight and fluorescent dye-conjugated IgG secondary antibodies as described above, followed by mounting with ProLong™ Glass Antifade Mountant (Catalog# P36980, Invitrogen). The images were acquired with Leica TCS SPE Confocal Microscope or Leica TCS SP8 Confocal Microscope using a ×63 oil immersion objective (University Life Science, PolyU). Z stacks of immunofluorescence staining were captured with a 0.4–0.5 μm separation between Z stack slices. Image deconvolution of confocal images were performed with Imaris (Oxford Instruments) or Lightning deconvolution (Leica).

Colocalization of (1) LC3 puncta and mitochondria; (2) LAMP1 puncta and mitochondria; (3) Rab5 puncta and mitochondria; (4) Rab5-LC3 puncta and mitochondria, and (5) Rab5-LAMP2A puncta and mitochondria were manually determined using single Z stacks slice that shows co-staining with green and red fluorescence. Colocalization coefficient of (1) p62 puncta and mitochondria and (2) TGN38 and NLRP3 puncta was measured by Coloc 2 plugin on the Fiji image processing platform. At least 50 cells in each group were subjected to quantification. The whiskers in the box and whisker plots extend to the minimum and maximum values, while the box presents the 25th and 75th percentiles with the central line at the median.

**Cytosolic fraction isolation, mitochondrial DNA quantification, and depletion.** Cytosolic fraction of BMDM was isolated using a mitochondria isolation kit according to manufacturers' instructions (Catalog# 89874, Thermo Scientific). DNA was isolated from cytosolic fraction using DNeasy Blood and Tissue Kit (Catalog# 69504, QIAGEN). QPCR was used to quantify mitochondrial DNA copy number using primers specific to *mtCO1* normalized with nuclear DNA encoding

*18 S ribosomal RNA*. For mitochondrial DNA depletion, BMDM were treated with 100 ng/ml ethidium bromide (Catalog# 15585011, Invitrogen) for 4 days before LPS priming and ATP treatment. Quantification of cytosolic oxidized mtDNA was performed by measuring the 8-OH-dG content in the cytosolic mtDNA using an 8-OH-dG ELISA kit (Catalog# ab201734, Abcam). One microgram of DNA isolated from the cytosolic fraction as described above was digested with nuclease P1 and dephosphorylated by alkaline phosphatase before the ELISA analysis.

**Cell viability assay**. Pierce LDH Cytotoxicity Assay Kit (Catalog# 88953, Thermo Scientific) was used according to manufacturer's instruction to measure cell viability after activation of NLRP3 inflammasome in BMDM. Briefly, $1 \times 10^4$ cells/well were seeded onto 96-well plate in FBS free DMEM. Cells were lysed to determine maximal lactate dehydrogenase (LDH) activity and PBS was added to cells to quantify spontaneous release of LDH. Viability of BMDM was defined as

$$\left( 1 - \frac{LPS + ATP - induced\ LDH\ release - Spontaneous\ LDH\ release}{Maximal\ LDH - Spontaneous\ LDH\ release} \right) \times 100.$$

**Statistical and reproducibility**. Data were analyzed with GraphPad Prism 6.0 or SPSS. Data were presented as mean ± SEM. Statistical significance between two group comparison was determined using unpaired Student's *t*-test or nonparametric Mann–Whitney U test. Statistical significance between multiple group comparison was determined using one-way ANOVA with Bonferroni correction or nonparametric Kruskal–Wallis test with Dunn's test. Equal variance and normality were determined by Levene's test and D'Agostino-Pearson omnibus normality test, respectively. A *p*-value less than 0.05 is considered statistically significant. In vitro experiments were repeated three times independently with similar results. Animal experiments were performed one time with each animal as a biologically independent sample.

**Reporting summary**. Further information on research design is available in the Nature Research Reporting Summary linked to this article.

## Data availability

The data generated in this study are provided in the Supplementary Information/Source Data file. Source data are provided with this paper.

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

## Acknowledgements

The imaging experiments were supported by Dr. Michael YF Yuen in University of Life Science, PolyU. We would like to thank Yawen Zhou from HKU for her help on bone marrow transplantation in mice, Miss HUNG Choi Hang from PolyU for her help on H&E staining in eWAT and Miss Chan Ka Ying for her help on quantification of imaging data and generating some of the data. This work is supported by Research Grants Council General Research Fund (171018/15M), National Natural Science Foundation of China (NSFC) (grant number: 91857119), and PolyU Internal Funding to K.K.Y.C.

## Author contributions

K.K.L.W. performed the experiments and drafted the manuscript. K.K.L. and H.G.L. performed the experiments. P.M.F.S. contributed reagents and advice on autophagy study. R.L.C.H. and A.X. reviewed manuscript and contributed to direction and experimental design. D.Y. reviewed manuscript and advised the study. K.K.Y.C. contributed to experimental design, funding seeking, supervision of the work and manuscript writing.

## Competing interests

The authors declare no competing interests.
