## [Peer Review File · Nature Communications]

The APPL1-Rab5 axis restricts NLRP3 inflammasome activation through early endosomal-dependent mitophagy in macrophagesREVIEWER COMMENTS

Reviewer #1 (Remarks to the Author):

It is known that removal of dysfunctional mitochondria via mitophagy restricts activation of the NLRP3 inflammasome. This study has identified a novel mechanism involving endosomal-mediated mitophagy as a mechanism to limit NLRP3 activation. Specifically, the authors report that both endosomal Tab5 and APPL1 co-localize with mitochondria to facilitate endosomal-mediated mitochondrial clearance. Mice with hematopoietic deletion of APPL gene are more sensitive to endotoxin-induced sepsis and obesity-induced inflammation possibly due to reduced mitophagy. These mice also had elevated systemic levels of interleukin-1beta, a major product of the NLRP3 inflammasome. Overall, these findings demonstrate that APPL/Rab5-mediated mitophagy is important in restricting inflammasome activation in macrophages. Although this is an interesting study, there are a number of concerns that need to be addressed by the authors. In particular, the relationship between endosomes and autophagosomes need to be further clarified. Better and higher quality methods to assess mitophagy are also needed. Detailed concerns are discussed below.

- A key question that has not been investigated by the authors is whether the defect in mitophagy leads to accumulation of mitochondria in APPL1 KO cells. That is, is there a difference in mitochondrial mass in the KO cells due to reduced turnover at baseline or is the defect only obvious after LPS+ATP treatment? The authors should assess mitochondrial mass in WT and KO cells using western blotting for various mitochondrial proteins (outer and inner/matrix proteins), as well as assessing mtDNA copy number or citrate synthase activity.

- Another major question that needs to be addressed is the relationship between Rab5/APPL-positive endosomes and LC3-positive autophagosomes in this study. The authors show that mitochondria accumulate in LC3-positive autophagosomes due to impaired fusion with lysosomes. Are these vesicles containing mitochondria then also positive for Rab5/APPL or are these a different set of vesicles involved in mitophagy? Is it possible that mitochondria are sequestered in two different types of vesicles (i.e. Rab5-positive endosomes and LC3II-positive autophagosomes)?

- Experiments assessing mitophagy in this study are not very sophisticated. In addition to assessing potential effect on baseline levels of mitophagy as discussed above, the authors need to perform additional experiments to strengthen the mitophagy data. First, the mitophagy experiments in Figure 7 and Suppl. Figure 4A+B should include an additional mitochondrial protein marker that is inside the mitochondria. It is possible that Tom20 in the outer membrane is subjected to proteasomal degradation during the stress without an effect on mitochondrial content. Also, the imaging experiment in Supplemental Figure 4C lack quantitation of co-localization.

- More importantly, the authors are using MitoTracker Deep Red to assess mitophagy in co-localization experiments presumably to detect mitochondria inside vesicles. There are several concerns with this experiment. First, as stated by the authors in Fig. 2C, MTDR staining of mitochondria is dependent on

membrane potential. Mitochondria that are subjected to mitophagy have usually lost their membrane potential. Thus, the authors need to address this discrepancy and the authors need use a different mitochondrial marker that is independent of membrane potential in their mitophagy experiments or use a pH sensitive mitophagy reporter such as Mito-Keima. Also, the quantification of co-localization events in Fig. 4B & 4D list n=15 and n=10. If this refers to the number of cells analyzed, this number is too low. Also, how many times were the experiments repeated?

- Experiment in Supplemental Figure 3 – the rationale for these experiment are unclear and lack important controls. The authors need to compare the change in membrane potential in WT and KO cells w/o ATP treatment. The current data are normalized to WT vehicle control which is not appropriate. The authors need to include the proper control or remove this figure.

- Western blot of whole cell lysates in Figure 1D is missing a loading control

- Typo - in “APPL1+ endosomes are recruited to mitochondria upon NLRP3 inflammasome.....” section. Authors refers to Fig 6A but should be 5A.

- Figure 5 – does n=10 refer to the number of cells analyzed for the number of Rab5+ positive vesicle and co-localization with MTR? The authors need to clearly indicate how many independent experiments were performed and how many cells were analyzed in each experiment per condition.

- Figure 7 – Imaging experiments - How times were the experiment repeated and many cells were analyzed in each condition?

Reviewer #2 (Remarks to the Author):

Endocytic trafficking and signaling form the reciprocal interconnectivity to regulate physiological and pathological function (Di Fiore et al. 2014 and Schmid 2017). APPL1 was identified as Akt-interacting protein and associates with Rab proteins and many receptors to modulate vesicle transport and receptor signaling. APPL1-positive endosomes represent a subpopulation of Rab5-positive endosomes and APPL1 proteins translocate to different endosomal and cellular compartments in response to different stimuli (Kalaidzidis et al. 2015, Miaczynska et al. 2004, Lakoduk et al. 2019). It's important to expand our understanding of the function of APPL1 in cell context.

In this manuscript, Wu et al. reported that endosomal APPL1 selectively regulates NLRP3 inflammasome via influence on mitophagy in macrophages. Unlike involvement in EGFR and insulin signaling, this work points out a new role of APPL1 in response to stress to facilitate mitophagy. Some data and interpretation should be further addressed to support the work.

Major points:

1. APPL1 involves in trafficking and signaling and form feedback loop with signaling. Is the effect of APPL1 in mitophagy, mitochondrial ROS, IL-1 β and inflammasome direct or indirect consequence in KO BMDM? Does APPL1 KO change the differentiation efficiency of macrophages and the characteristics of BMDM?
2. Only colocalization of IF data cannot fully support the recruitment of APPL1-positive endosomes are to mitochondria. LPS+ATP treatment and APPL1 KO create big aggregates in IF data especially in perinuclear area (Fig.4, 5, 7, suppl. Figs.). Some experiments like fractionation should be used to support the recruitment of APPL1-positive endosomes are to mitochondria.
3. Some of 'representative' WB data (Fig. 1D, 3E, suppl. Fig2A) should be improved to have 'publishable' quality to support the story and represent the quantification result.
4. In Fig.7, Tom 20 was used as a measurement mitophagy. However, it was not explained and interpreted in the text.

Minor points:

1. Miss-labeling. On page 10, it describes " ...and mitochondria upon NLRP3 agonist stimulation (Figure 6A... ". It should be Figure 5A.
2. In suppl. Fig.7, it's not convinced that APPL1 "largely" co-localized with Rab5...

Reviewer #3 (Remarks to the Author):

Wu and colleagues study the effect of APPL1 deficiency on NLRP3 activation both in vitro and in vivo. The document a role for endosomal Rab5 regulating mitophagy, without which, excess mtDNA promotes NLRP3 activation. I have major concerns about the validity of some approaches and conclusions, but overall it is clear that the authors have put a lot of work into this manuscript.

Major points:

The data in the literature arguing for direct activation of NLRP3 by mitochondrial DNA is weak. It is more reasonable to assume that if mtDNA is accumulating in the cytosol then it could directly trigger AIM2 inflammasome activation, or that it will trigger cGAS/STING, and this could in turn activate the NLRP3 inflammasome. Both of these possibilities could be tested.

Again regarding the mechanism for NLRP3 activation, the most compelling data recently shows that it needs to accumulate on a damaged trans-golgi network. It seems most likely that this is what APPL1 deficiency is promoting to activate NLRP3, not mtDNA release. Some imaging or other investigation of this would be informative.

It is confusing to me why the baseline increase in mitochondrial damage does not spontaneously activate the NLRP3 inflammasome in APPL1 deficient cells without stimulation?

Specificity for NLRP3 is demonstrated by treatment with flagellin or MDP which show no difference for APPL1 deficient cells. However, it is unclear exactly how those experiments were performed (transfection?) and at any rate MDP is a Nod2 agonist that does not activate NLRP1. More specific protocols for non-NLRP3 activation should be performed.

Regarding mouse models, I would not necessarily consider either of these a definitive NLRP3 test, but they have been associated with NLRP3 and so may suffice. For the high-dose LPS challenge, this should presumably trigger Casp11 noncanonical inflammasome and subsequent NLRP3 activation, which is why they see an augmented response? Perhaps the authors could formally test transfected LPS in their APPL1 ko cell lines to confirm this.

Point-by-point replies to the reviewers

(NCOMMS-20-26496-T, by Wu KKL et al, entitled “An early endosomal-dependent mitophagy mediated by APPL1-Rab5 axis restricts NLRP3 inflammasome activation in macrophages”)

First of all, we would like to express our sincere thanks to the editor and reviewers for their constructive comments. We have now revised our manuscript according to your comments. Our point-by-point responses are listed below.

Reviewer #1 (Remarks to the Author):

It is known that removal of dysfunctional mitochondria via mitophagy restricts activation of the NLRP3 inflammasome. This study has identified a novel mechanism involving endosomal-mediated mitophagy as a mechanism to limit NLRP3 activation. Specifically, the authors report that both endosomal Tab5 and APPL1 co-localize with mitochondria to facilitate endosomal-mediated mitochondrial clearance. Mice with hematopoietic deletion of APPL gene are more sensitive to endotoxin-induced sepsis and obesity-induced inflammation possibly due to reduced mitophagy. These mice also had elevated systemic levels of interleukin-1beta, a major product of the NLRP3 inflammasome. Overall, these findings demonstrate that APPL/Rab5-mediated mitophagy is important in restricting inflammasome activation in macrophages. Although this is an interesting study, there are a number of concerns that need to be addressed by the authors. In particular, the relationship between endosomes and autophagosomes need to be further clarified. Better and higher quality methods to assess mitophagy are also needed. Detailed concerns are discussed below.

Response: Thank you for your constructive suggestion for different experiments on mitophagy assessment and the relationship between Rab5⁺/APPL1⁺ endosomes and autophagosomes. We have now used Mito-Keima to monitor mitophagy flux (*Fig. 4B, Fig. 7C, Supplementary 5B and Supplementary 6*) and subcellular fractionation approach to assess the recruitment of APPL1⁺/Rab5⁺ endosomes and p62 (another indicator of mitophagy) to the mitochondria in response to inflammasome activation (*Fig. 5B and Supplementary Fig. 8A*). In addition, we have performed additional triple immunofluorescence staining to visualize the trafficking of APPL1⁺/Rab5⁺ endosomes to the mitochondria, mitophagosomes and mitolysosomes in BMDM upon inflammasome activation (*Supplementary Fig. 9A-B and 10A-B*). Please find our detailed response below.

Q1. A key question that has not been investigate by the authors is whether the defect in mitophagy leads to accumulation of mitochondria in APPL1 KO cells. That is, is there a difference in mitochondrial mass in the KO cells due to reduced turnover at baseline or is the defect only obvious after LPS+ATP treatment? The authors should assess mitochondrial mass in WT and KO cells using western blotting for various mitochondrial proteins (outer and inner/matrix proteins), as well as assessing mtDNA copy number or citrate synthase activity.

Response: We have now measured mitochondrial mass in the LPS-treated BMDM. Immunoblotting analysis indicated that protein expression of cytochrome C (Cyto C; expresses and localizes at the inner membrane of mitochondria), Tom20 (the outer protein of mitochondria) and pyruvate dehydrogenase (the matrix protein of mitochondria) expressed similarly in WT BMDM and KO BMDM after LPS treatment for 20 hours (*Supplementary Fig. 2E*). Consistently, mtDNA copy number was also similar between the two genotypes (*Supplementary Fig. 2D; line 11-15, page 7*). In addition, Mito-Keima experiment showed that mitophagy are comparable between APPL1 KO and WT BMDM under LPS alone condition (*Fig. 4B and Supplementary Fig. 5B; line 14-16, page 10*). Therefore, we believe that APPL1 deficiency causes defective mitophagy under inflammasome activation condition only.

Q2. Another major question that needs to be addressed is the relationship between Rab5/APPL-positive endosomes and LC3-positive autophagosomes in this study. The authors show that mitochondria accumulate in LC3-positive autophagosomes due to impaired fusion with lysosomes. Are these vesicles containing mitochondria then also positive for Rab5/APPL or are these a different set of vesicles involved in mitophagy? Is it possible that mitochondria are sequestered in two different types of vesicles (i.e. Rab5-positive endosomes and LC3II-positive autophagosomes)?

Response: Triple immunofluorescence staining revealed that Rab5⁺ and APPL1⁺ signals are co-localized in BMDM under LPS priming condition, and LPS+ATP stimulation further provoked the co-localization in a time-dependent manner (*Supplementary Fig. 9A*). In addition, LPS+ATP stimulation induced the recruitment of APPL1⁺ and Rab5⁺ endosomes to the mitochondria (*Fig. 5C and Supplementary Fig. 9A&B & 10A*). Co-localization of APPL1⁺/Rab5⁺ endosomes with mitochondria and LC3⁺ puncta (known as mitophagosome; MitoTracker⁺LC3II⁺) increased upon LPS+ATP treatment (*Supplementary Fig. 9B and 10A*). On the other hand, very little number of APPL1⁺/Rab5⁺ endosomes was found to be co-localized with mitochondria and lysosome (known as mitochondrial autolysosomes; MitoTracker⁺LAMP1⁺ in *Supplementary Fig. 9C* and MitoTracker⁺LAMP2A⁺ in *Supplementary Fig. 10H*) (usually less than one or even zero triple positive signal per cell). Consistent to our previous findings, APPL1 deficiency led to increased accumulation of mitophagosome (MitoTracker⁺ and LC3⁺) (*Supplementary Fig. 10A & G*) and decreased number of Rab5⁺ puncta on the mitochondria (*Supplementary Fig. 10A & F*). The increased number of mitophagosomes in APPL1 KO BMDM was mainly from Rab5⁻ mitophagosomes, whereas the number of Rab5⁺ mitophagosomes was comparable between APPL1 KO and WT BMDM under LPS+ATP condition (*Supplementary Fig. 10A & B-C*). However, when the data is presented as percentage of total mitophagosome population, we found that the percentage of Rab5⁺ mitophagosomes was significantly lower in APPL1 KO BMDM than that in WT BMDM under LPS+ATP condition (*Supplementary Fig. 10A & D*).

Taken together, the above data suggest that there are at least two sets of vesicles recruited to mitophagosomes (i.e. Rab5⁺ and Rab5⁻ endosomes or APPL1⁺ and APPL1⁻ endosomes) upon inflammasome activation, yet further investigation of their biochemical characteristics and regulatory network by APPL1 using live imaging is warrant.

Q3. Experiments assessing mitophagy in this study are not very sophisticated. In addition to assessing potential effect on baseline levels of mitophagy as discussed above, the authors need to perform additional experiments to strengthen the mitophagy data. First, the mitophagy experiments in Figure 7 and Suppl. Figure 4A+B should include an additional mitochondrial

protein marker that is inside the mitochondria. It is possible that Tom20 in the outer membrane is subjected to proteasomal degradation during the stress without an effect on mitochondrial content. Also, the imaging experiment in Supplemental Figure 4C lack quantitation of co-localization.

Q4. More importantly, the authors are using MitoTracker Deep Red to assess mitophagy in co-localization experiments presumably to detect mitochondria inside vesicles. There are several concerns with this experiment. First, as stated by the authors in Fig. 2C, MTDR staining of mitochondria is dependent on membrane potential. Mitochondria that are subjected to mitophagy have usually lost their membrane potential. Thus, the authors need to address this discrepancy and the authors need use a different mitochondrial marker that is independent of membrane potential in their mitophagy experiments or use a pH sensitive mitophagy reporter such as Mito-Keima. Also, the quantification of co-localization events in Fig. 4B & 4D list n=15 and n=10. If this refers to the number of cells analyzed, this number is too low. Also, how many times were the experiments repeated?

Response to Q3 and Q4: Thank you for your suggestion.

We have now used multiple approaches to assess mitophagy as described as below:

1) For the concerns about Figure 7 (*Fig. 7A and Supplementary 5C in the revised manuscript*) and Suppl. Figure 4A+B (*Fig. 4A and Supplementary Fig. 5A in the revised manuscript*), we have now included the inner membrane protein of mitochondria, cytochrome c, as mitochondrial marker and clearance. Consistent to the change in Tom20 expression, inflammasome stimuli (LPS+ATP or LPS+FCCP) reduced expression of cytochrome C in WT BMDM, whereas genetic ablation of APPL1 abolished the reduction (*Fig. 4 and Fig. 7*). Replenishment of wild-type APPL1 or overexpression of Rab5-Q79L reversed the defective mitophagy in APPL1 KO BMDM (*Supplementary Fig. 5C*). On the contrary, replenishment of APPL1-Triple or APPL1-N308D mutant was unable to restore the defect (*Supplementary Fig. 5C*).

2) We have also followed the Reviewer's suggestion to employ Mito-Keima for quantitative measurement of mitophagy. Briefly, WT and APPL1 KO BMDM were infected with adenovirus encoding Mito-Keima for 24 hours, followed by priming with LPS for 20 hours and subsequent ATP or FCCP stimulation for indicated time points. Please refer to the Methodology of our manuscript for the detailed experimental procedure (*Line 3-11, Page 26*). Flow cytometry analysis showed that WT and APPL1 KO BMDM exhibited similar mitophagy under LPS alone condition, as reflected by a similar percentage of the cells with a high 561/405 nm ratio from Mito-Keima. As expected, FCCP or ATP markedly increased the percentage of cells with the high 561/405 nM ratio in WT BMDM (from 6.154 % to 16.757% and 10.097% to 20.363%, respectively), whereas APPL1 deficiency significantly diminished these mitophagic responses (only increase from 6.377% to 10.694% and 9.594 to 14.125%) (*Fig. 4B, Supplementary Fig. 5B and Supplementary Fig. 6; line 14-20, page 10*).

By using Mito-Keima mitophagy assay, we have assessed whether the promoting effect of APPL1 on mitophagy requires its endosomal localization and Rab5 binding ability. Consistent

with our imaging and immunoblotting data, wild-type APPL1 but not its mutants with defective endosomal localization (APPL1-Triple) or Rab5 binding (APPL1-N308D) was able to rescue the impairment of mitophagy in APPL1 KO BMDM (*Fig. 7C*). In addition, activation of Rab5 by Rab5-Q79L mutant also rescued the defective mitophagy induced by APPL1 deficiency (*Fig. 7C; line 15-20, page 14*).

3) We have also monitored the mitophagy response using the subcellular fractionation method as suggested by Reviewer 2 (Please refer our response to Q2 to Reviewer 2). In brief, we showed that recruitment of the mitophagy regulator p62 onto the mitochondria induced by LPS+ATP was diminished by APPL1 deficiency (*Supplementary Fig. 8A*).

The above new data together with the original immunofluorescence staining data (*Fig. 4C*) support the notion that APPL1 is essential for mitophagy in response to inflammasome activation.

For the concern about *Supplementary Fig. 4C* (*Supplementary Fig. 7A in the revised manuscript*), we have now quantified the number of p62 puncta on mitochondria (labelled by MitoTracker). Our quantification analysis indicated that treatment with LPS+ATP dramatically induced the formation of p62 puncta on the mitochondria in WT BMDM, but such effect of LPS+ATP was diminished by APPL1 deficiency (*Supplementary Fig. 7A*). In addition, the subcellular fractionation experiment also showed a decrease of p62 expression in the mitochondrial fraction in APPL1 KO BMDM when compared to WT BMDM under LPS+ATP condition, as mentioned above (*Supplementary Fig. 8A; line 18-21, page 11*).

For the concern of sample size, we have quantified at least 50 cells in all the imaging experiments (*Fig. 4C, Fig. 5D, Fig. 7B, Supplementary Fig. 7 and Supplementary Fig. 10*).

For the concern of the use of MitoTracker Deep Red in the assessment of mitophagy, we have used Mito-Keima mitophagy assay to quantitatively monitor mitophagy as stated above (*Fig. 4C-D, Fig. 5D, Fig. 7B, Supplementary Fig. 7 and Supplementary Fig. 9 and Supplementary Fig. 10*).

Q5. Experiment in Supplemental Figure 3 – the rationale for these experiment are unclear and lack important controls. The authors need to compare the change in membrane potential in WT and KO cells w/o ATP treatment. The current data are normalized to WT vehicle control which is not appropriate. The authors need to include the proper control or remove this figure.

- 1) *Supplementary Figure 3D* should change to LPS alone as control.
- 2) Western blot of whole cell lysates in *Figure 1D* is missing a loading control

Response: In *Figure 1 and 2*, we showed that APPL1 deficiency exacerbates LPS+ATP-induced inflammasome activation and mitochondrial damages. Apart from mitochondrial damages, ATP is known to induce inflammasome activation via multiple pathways such as disrupting intracellular Ca^{2+} and K^{+} balance. To further investigate whether APPL1 specifically modulates mitochondrial stress-induced inflammasome activation, we therefore treated the LPS-primed BMDM with various mitochondrial-targeted compounds including FCCP (a

mitochondrial oxidative phosphorylation uncoupler), antimycin A (the complex III inhibitor) and rotenone (the complex I inhibitor) as previously described¹. We have now included a more detailed description in the main text (*Please refer to line 15-18 page 7*).

1) In Figure Supplementary Fig. 3D, we have now repeated the experiment and normalized all the data with WT+LPS alone.

2) In Fig. 1D, we have included HSP90 as a loading control.

- Typo - in “APPL1+ endosomes are recruited to mitochondria upon NLRP3 inflammasome.....” section. Authors refers to Fig 6A but should be 5A.

Response: Thank you, we have revised accordingly.

- Figure 5 – does n=10 refer to the number of cells analyzed for the number of Rab5+ positive vesicle and co-localization with MTR? The authors need to clearly indicate how many independent experiments were performed and how many cells were analyzed in each experiment per condition.

Response: We performed at least 2-3 times independent experiments. We have included at least 50 cells in the imaging experiments (*Fig. 4C-D, Fig. 5D, Fig. 7B, Supplementary Fig. 7 and Supplementary Fig. 9 and Supplementary Fig. 10*).

Figure 7 – Imaging experiments - How times were the experiment repeated and many cells were analyzed in each condition?

Response: The experiments have been performed at least 2-3 times and 50 cells were counted in each experiment. Representative images and quantification data were shown in the Figures.

Reviewer #2 (Remarks to the Author):

Endocytic trafficking and signaling form the reciprocal interconnectivity to regulate physiological and pathological function (Di Fiore et al. 2014 and Schmid 2017). APPL1 was identified as Akt-interacting protein and associates with Rab proteins and many receptors to modulate vesicle transport and receptor signaling. APPL1-positive endosomes represent a subpopulation of Rab5-positive endosomes and APPL1 proteins translocate to different endosomal and cellular compartments in response to different stimuli (Kalaidzidis et al. 2015, Miaczynska et al. 2004, Lakoduk et al. 2019). It's important to expand our understanding of the function of APPL1 in cell context.

In this manuscript, Wu et al. reported that endosomal APPL1 selectively regulates NLRP3 inflammasome via influence on mitophagy in macrophages. Unlike involvement in EGFR and insulin signaling, this work points out a new role of APPL1 in response to stress to facilitate mitophagy. Some data and interpretation should be further addressed to support the work.

Major points:

Q1. APPL1 involves in trafficking and signaling and form feedback loop with signaling. Is the effect of APPL1 in mitophagy, mitochondrial ROS, IL-1 β and inflammasome direct or indirect consequence in KO BMDM? Does APPL1 KO change the differentiation efficiency of macrophages and the characteristics of BMDM?

Response:

NF- κ B activation induced by LPS is the priming step, with a tight control of signaling transduction and feedback mechanism. Although previous studies demonstrated that APPL1 differentially regulates NF- κ B activity in different cell types (such as pancreatic β cells and HEK293 cells)^{2,3}, we did not observe any impact of APPL1 deficiency on LPS-induced NF- κ B activation in macrophages (*Supplementary Fig. 2B*). Instead, we provided multiple lines of evidence showing that APPL1 promotes mitophagic flux via facilitating Rab5⁺ early endosomes onto the damaged mitochondria, which in turn diminishes the production of mitochondria danger signals, and hence restricting NLRP3 inflammasome activation and IL-1 β production in macrophages (*Fig. 6, Fig. 7, and Supplementary Fig. 5 and Supplementary Fig. 8*). Our rescue experiments in APPL1 KO BMDM revealed that the regulatory actions of APPL1 on Rab5-mediated mitophagy and NLRP3 requires the binding ability of APPL1 with Rab5 and the endosomal localization of APPL1. Previous studies also demonstrated that APPL1 directly interacts with Rab5 on the early endosome and controls its activity^{4,5}. Taken together, we believe that APPL1 directly controls mitophagy via Rab5, which restricts NLRP3 inflammasome activation.

To determine the effect of APPL1 deficiency on macrophage differentiation, we measured the surrogate macrophage markers (F4/80 and CD11b) in BMDM by flow cytometry. Upon stimulation with the differentiation medium from L929 cells for 7 days, a similar percentage of cells from APPL1 KO mice and WT controls were differentiated into F4/80 and CD11b double positive cells (*Supplementary Fig. 2A*).

Q2. Only colocalization of IF data cannot fully support the recruitment of APPL1-positive endosomes are to mitochondria. LPS+ATP treatment and APPL1 KO create big aggregates in

IF data especially in perinuclear area (Fig.4, 5, 7, suppl. Figs.). Some experiments like fractionation should be used to support the recruitment of APPL1-positive endosomes are to mitochondria.

Response: Thank you for your suggestion. As replied to Q3 and Q4 to Reviewer 1, we have determined whether APPL1⁺ endosomes are recruited to the mitochondria upon LPS+ATP treatment using the subcellular fractionation method. After treatment with LPS+ATP for 0, 5 and 15 minutes, WT BMDM were collected and subjected to isolation of mitochondria using a Mitochondria Isolation Kit for Culture Cells (Cat. No. 89874, ThermoFisher Scientific Inc.). Immunoblotting analysis revealed that the mitochondrial matrix protein cytochrome C but not GAPDH was detected in the mitochondrial fraction, validating the methodology we used (*Please see below diagram*).

BMDM were isolated from C57BL/6J mice and subjected to LPS priming (100 ng/ml) for 20 hours, followed by ATP stimulation (3 mM) as indicated. The cells were then fractionated using Mitochondria Isolation Kit for Cultured Cells (Catalog# 89874, Thermofisher) to obtain the mitochondrial and cytosolic fractions. Lysate from each fraction was subjected to immunoblotting analysis of Cytochrome C (Cyto C; mitochondrial marker) and GAPDH (cytosolic protein) to confirm the successful isolation of mitochondrial and cytosolic fraction respectively. M: Protein ladder (Catalog# 26617, Thermofisher), the number represents the molecular weight of the protein marker in kDa.

Consistent to our immunofluorescence data and the previous study⁶, expressions of APPL1, Rab5 and p62 were increased in the mitochondrial fractions upon LPS+ATP treatment in BMDM (*Fig. 5B*).

We have also employed the above fractionation method to confirm that:

(1) The recruitment of APPL1 to the mitochondria induced by LPS+ATP was diminished when the endosomal localization or Rab5 binding ability of APPL1 was blocked (*Supplementary Fig. 8A*).

(2) APPL1 deficiency abrogated the recruitment of Rab5⁺ endosomes and p62 to the mitochondria upon LPS+ATP treatment, while replenishment of wild-type APPL1 and constitutively active Rab5-Q79L mutant but not the APPL1-Triple or APPL1 N308D mutant reversed the abrogation in APPL1 KO BMDM (*Supplementary Fig. 8A*).

All the above data are consistent to our immunofluorescence staining data shown in Fig. 5C-D, Fig. 7B and Supplementary Fig. 7A.

(3) For the concern of Fig. 4 in which examining whether APPL1 deficiency affects mitophagic flux, we have used Mito-Keima to quantitatively monitor mitophagy. Consistent to the immunofluorescent data, we found that APPL1 deficiency diminished mitophagy flux induced by LPS+ATP or LPS+FCCP (*Fig. 4B, Fig. 7C and Supplementary Fig. 5B&6*). In addition, replenishment of wild-type APPL1 but not its Triple or N308D mutant rescued the defects in

mitophagy flux in APPL1 KO BMDM (*Fig. 7C*). Activation of Rab5 was also able to restore mitophagy flux in APPL1 KO BMDM (*Fig. 7C*)

Q3. Some of ‘representative’ WB data (Fig. 1D, 3E, suppl. Fig2A) should be improved to have ‘publishable’ quality to support the story and represent the quantification result.

Response: We have re-run the Western Blot for Fig. 1D, Fig. 3E and Supplementary Fig. 2A (Supplementary Fig. 2B in the revised manuscript).

4. In Fig.7, Tom 20 was used as a measurement mitophagy. However, it was not explained and interpreted in the text.

Response: Tom20 is a mitochondrial import receptor subunit that localizes on the outer mitochondrial membrane. It is commonly used as a mitochondria marker. The degradation of Tom20 and cytochrome C (a mitochondrial inner membrane protein) reflect the mitochondrial clearance and mitophagic rate in the BMDM^{7,8}. We have now explained this in the text (*please refer to line 23, page 9 and line 1-3, page 10*).

Minor points:

1. Miss-labeling. On page 10, it describes “...and mitochondria upon NLRP3 agonist stimulation (Figure 6A... “. It should be Figure 5A.
2. In suppl. Fig.7, it’s not convinced that APPL1 “largely” co-localized with Rab5.

Response:

1. We have confirmed that all figures are correctly numbered.
2. We change the statement to “Confocal images indicated that a portion of full-length wild-type APPL1 co-localized with Rab5 and effectively recruited to the mitochondria.....”. *Please refer to line 25 page12*

Reviewer #3 (Remarks to the Author):

Wu and colleagues study the effect of APPL1 deficiency on NLRP3 activation both in vitro and in vivo. The document a role for endosomal Rab5 regulating mitophagy, without which, excess mtDNA promotes NLRP3 activation. I have major concerns about the validity of some approaches and conclusions, but overall it is clear that the authors have put a lot of work into this manuscript.

Major points:

Q1. The data in the literature arguing for direct activation of NLRP3 by mitochondrial DNA is weak. It is more reasonable to assume that if mtDNA is accumulating in the cytosol then it

could directly trigger AIM2 inflammasome activation, or that it will trigger cGAS/STING, and this could in turn activate the NLRP3 inflammasome. Both of these possibilities could be tested.

Response: To address the above possibilities, we have now performed the following experiments:

1. We first tested whether APPL1 deficiency has any impact on cytosolic DNA-induced IL-1 β production. To this end, APPL1 KO and WT BMDM were primed with LPS (100 ng/ml) for 20 hours, followed by stimulation with the synthetic B-form double-stranded Poly(dA:dT) (which utilizes AIM2) for 6 hours. Similar IL-1 β secretion was detected in APPL1 KO and WT BMDM upon stimulation with LPS+Poly (dA:dT) (*Supplementary Fig. 1F; line 25 page 5-line1-2, page 6*), suggesting that APPL1 does not regulate AIM2 inflammasome activation.

2. To explore whether AIM2 or STING mediates APPL1 deficiency-induced inflammasome activation, we transfected WT or APPL1 KO BMDM with siRNA against *AIM2* or *STING* using DharmaFECT3 transfection reagents (Catalog #: T-2003-03, Horizon), which has been shown to downregulate the target gene more than 80% with minimal cytotoxicity⁹. QPCR analysis and/or immunoblotting confirmed the downregulation of AIM2 or STING in WT and APPL1 KO BMDM after the transfection for 48 hours, accompanied with diminished IL-1 β production under LPS+ Poly (dA:dT) stimulation condition. By using the same gene silencing approach, we found that neither knockdown of *AIM2* or *STING* expression abolished APPL1 deficiency-induced NLRP3 inflammasome activation under LPS+ATP or LPS+FCCP condition (*Supplementary Fig. 4A-E; line 5-11, page 9*).

3. We also tested whether pharmacological inhibition of cGAS attenuates APPL1 deficiency-induced NLRP3 inflammasome activation. Treatment with the cGAS inhibitor RU.521¹⁰ partially reduced IL-1 β production in APPL1 KO BMDM upon LPS+ATP stimulation, when compared to those treated with vehicle (*Supplementary Fig. 4F; line 11-17, page 9*). Unlike the rescue effects by treatment with NAC, MitoTEMPO or ethidium bromide, IL-1 β production remained significantly higher in APPL1 KO BMDM when compared to WT BMDM under RU.521 treatment condition.

4. Recent studies demonstrated that oxidized mtDNA induced inflammasome activation via NLRP3 but not in an AIM2 dependent manner^{11, 12}. Inhibiting the binding of oxidized mtDNA to NLRP3 using 8-hydroxy-guanosine (8-OH-dG; 200 μ M) completely abolished APPL1 deficiency-induced inflammasome activation in BMDM under LPS+ATP condition (*Fig. 3I-J; line 20-25, page 8 and line 1, page 9*). Furthermore, we found that the amount of oxidized mtDNA in the cytosol of APPL1 KO BMDM was much higher than that in WT BMDM (*Fig. 2E and 3G; line 9-11, page 7*). These data indicate that oxidized mtDNA might be the major contributor to the augmented inflammasome activation in APPL1 deficiency BMDM, although further investigation is required (for instance, compare whether oxidized mt-DNA-induced IL-1 β is higher in APPL1 KO BMDM than that in WT BMDM).

5. Our previous findings demonstrated that removal of mtDNA by ethidium bromide or inhibition of mitochondrial ROS by MitoTEMPO completely abrogated the potentiating effect of APPL1 deficiency on inflammasome activation under LPS+ATP condition (*Fig. 3*).

Together with the new findings described above, we believe that ROS and oxidized mtDNA derived from damaged mitochondria are the major contributors to the augmented inflammasome activation in APPL1 KO BMDM.

Q2. Again regarding the mechanism for NLRP3 activation, the most compelling data recently shows that it needs to accumulate on a damaged trans-golgi network. It seems most likely that this is what APPL1 deficiency is promoting to activate NLRP3, not mtDNA release. Some imaging or other investigation of this would be informative.

Response: Thank you for your suggestion. We have now determined whether APPL1 deficiency leads to accumulation of NLRP3 on the damaged trans-Golgi network under LPS+ATP conditions. Consistent to the previous study¹³, trans-Golgi (visualized by TGN38) was disrupted and the co-localization of TGN38 and NLRP3 was increased in similar magnitude in WT BMDM and APPL1 KO BMDM upon LPS+ATP stimulation (*Supplementary Fig. 7B; line 19-23, page 11*), suggesting APPL1 does not regulate this newly identified inflammasome related pathway.

Q3. It is confusing to me why the baseline increase in mitochondrial damage does not spontaneously active the NLRP3 inflammasome in APPL1 deficient cells without stimulation?

Response: Both the priming (mainly via the activation of NF- κ B and subsequent upregulation of pro-IL-1 β and NLRP3 expression) and the activation (such as mitochondrial damage, ATP and etc) steps are essential for full activation of NLRP3 inflammasome in macrophages¹⁴. Although APPL1 deficiency leads to mitochondrial dysfunction such as increase production of mtROS and increased number of damaged mitochondria under LPS condition (so-called baseline), it does not have any impact on the priming step (reflected by similar NF- κ B activity and pro-IL-1 β and NLRP3 expression-*Supplementary Fig. 2A*). Since the priming step is pre-required for robust NLRP3 activation^{14, 15}, we believe that mitochondrial damage alone is not sufficient to induce full NLRP3 activation in APPL1 KO BMDM. Similar to our study, knockdown of p62 leads to increased mitochondrial ROS production at the baseline (without any treatment), but it does not spontaneously induce IL-1 β production in BMDM (Please refer to Figure S2A and S1C-D, respectively, in reference⁶). Likewise, deletion of the autophagy regulator LC3B or beclin1 leads to modest increase in mtROS but does not spontaneously induce NLRP3 inflammasome activation (Figure 2C and Figure 1, respectively- in reference¹⁶). In addition, mitophagy remains intact in APPL1 KO BMDM under LPS-priming condition (*Fig. 4B and Supplementary Fig. 5B*), thus it might be still able to restrict NLRP3 inflammasome activation.

Q4. Specificity for NLRP3 is demonstrated by treatment with flagellin or MDP which show no difference for APPL1 deficient cells. However, it is unclear exactly how those experiments were performed (transfection?) and at any rate. MDP is a Nod2 agonist that does not activate NLRP1. More specific protocols for non-NLRP3 activation should be performed.

Response: Flagellin stimulation was performed via transfection. BMDM were first primed with LPS (100 ng/ml) for 24 hours, followed by the transfection with flagellin (20 µg/ml) for 8 hours using Lipofectamine 3000. Please find the details of transfection in Methodology (*Line 21-23, page 23*).

The NOD2 agonist Muramyl dipeptide (MDP) has been proposed to mediate NLRP1 inflammasome activation by promoting the formation of a NOD2-NLRP1-Caspase-1 complex¹⁷. Various studies have used MDP as NLRP1 inflammasome activator for mouse BMDM and human neurons¹⁸⁻²⁰. However, another study also demonstrated that MDP was unable to stimulate the NLRP1 inflammasome activation, and its effect on inflammasome is instead mediated by NLRP3²¹. To avoid controversial conclusion, we have now removed the data related to MDP from our manuscript. If the reviewers think that the effect of APPL1 deficiency on NLRP1 inflammasome is necessary, we are willing to further investigate whether anthrax lethal factor (a direct NLRP1 agonist²²)-induced IL-1β production differs between APPL1 KO and WT BMDM.

Q5. Regarding mouse models, I would not necessarily consider either of these a definitive NLRP3 test, but they have been associated with NLRP3 and so may suffice. For the high-dose LPS challenge, this should presumably trigger Casp11 noncanonical inflammasome and subsequent NLRP3 activation, which is why they see an augmented response? Perhaps the authors could formally test transfected LPS in their APPL1 ko cell lines to confirm this.

Response: Thank you for your suggestion. We have now transfected APPL1 KO and WT BMDM with LPS, followed by measurement of IL-1β in the conditional medium. No difference in IL-1β secretion among the two genotypes was found (*Supplementary Fig. 1G; line 24-25 page 5 to line 1-3 page 6*), suggesting that APPL1 appears not involving in Casp11 non-canonical inflammasome activation.

References:

1. Zhou, R., Yazdi, A.S., Menu, P. & Tschopp, J. A role for mitochondria in NLRP3 inflammasome activation. *Nature* **469**, 221-225 (2011).
2. Jiang, X. *et al.* APPL1 prevents pancreatic beta cell death and inflammation by dampening NF-kappaB activation in a mouse model of type 1 diabetes. *Diabetologia* **60**, 464-474 (2017).
3. Hupalowska, A., Pyrzynska, B. & Miaczynska, M. APPL1 regulates basal NF-kappaB activity by stabilizing NIK. *J Cell Sci* **125**, 4090-4102 (2012).
4. Miaczynska, M. *et al.* APPL proteins link Rab5 to nuclear signal transduction via an endosomal compartment. *Cell* **116**, 445-456 (2004).
5. Kim, S. *et al.* Evidence that the rab5 effector APPL1 mediates APP-betaCTF-induced dysfunction of endosomes in Down syndrome and Alzheimer's disease. *Mol Psychiatry* **21**, 707-716 (2016).
6. Zhong, Z. *et al.* NF-kappaB Restricts Inflammasome Activation via Elimination of Damaged Mitochondria. *Cell* **164**, 896-910 (2016).
7. Vives-Bauza, C. *et al.* PINK1-dependent recruitment of Parkin to mitochondria in mitophagy. *Proc Natl Acad Sci U S A* **107**, 378-383 (2010).
8. Geisler, S. *et al.* PINK1/Parkin-mediated mitophagy is dependent on VDAC1 and p62/SQSTM1. *Nat Cell Biol* **12**, 119-131 (2010).
9. Dong, S.X.M. *et al.* Transfection of hard-to-transfect primary human macrophages with Bax siRNA to reverse Resveratrol-induced apoptosis. *RNA Biol* **17**, 755-764 (2020).
10. Vincent, J. *et al.* Small molecule inhibition of cGAS reduces interferon expression in primary macrophages from autoimmune mice. *Nat Commun* **8**, 750 (2017).
11. Zhong, Z. *et al.* New mitochondrial DNA synthesis enables NLRP3 inflammasome activation. *Nature* **560**, 198-203 (2018).
12. Shimada, K. *et al.* Oxidized mitochondrial DNA activates the NLRP3 inflammasome during apoptosis. *Immunity* **36**, 401-414 (2012).
13. Chen, J. & Chen, Z.J. PtdIns4P on dispersed trans-Golgi network mediates NLRP3 inflammasome activation. *Nature* **564**, 71-76 (2018).
14. Wu, K.K., Cheung, S.W. & Cheng, K.K. NLRP3 Inflammasome Activation in Adipose Tissues and Its Implications on Metabolic Diseases. *Int J Mol Sci* **21** (2020).
15. Bauernfeind, F.G. *et al.* Cutting edge: NF-kappaB activating pattern recognition and cytokine receptors license NLRP3 inflammasome activation by regulating NLRP3 expression. *J Immunol* **183**, 787-791 (2009).
16. Nakahira, K. *et al.* Autophagy proteins regulate innate immune responses by inhibiting the release of mitochondrial DNA mediated by the NALP3 inflammasome. *Nat Immunol* **12**, 222-230 (2011).
17. Hsu, L.C. *et al.* A NOD2-NALP1 complex mediates caspase-1-dependent IL-1beta secretion in response to Bacillus anthracis infection and muramyl dipeptide. *Proc Natl Acad Sci U S A* **105**, 7803-7808 (2008).
18. Kaushal, V. *et al.* Neuronal NLRP1 inflammasome activation of Caspase-1 coordinately regulates inflammatory interleukin-1-beta production and axonal degeneration-associated Caspase-6 activation. *Cell Death Differ* **22**, 1676-1686 (2015).
19. Wong, W.T. *et al.* Repositioning of the beta-Blocker Carvedilol as a Novel Autophagy Inducer That Inhibits the NLRP3 Inflammasome. *Front Immunol* **9**, 1920 (2018).
20. Cimen, I. *et al.* Prevention of atherosclerosis by bioactive palmitoleate through suppression of organelle stress and inflammasome activation. *Sci Transl Med* **8**, 358ra126 (2016).
21. Kovarova, M. *et al.* NLRP1-dependent pyroptosis leads to acute lung injury and morbidity in mice. *J Immunol* **189**, 2006-2016 (2012).
22. Boyden, E.D. & Dietrich, W.F. Nalp1b controls mouse macrophage susceptibility to anthrax lethal toxin. *Nat Genet* **38**, 240-244 (2006).

REVIEWERS' COMMENTS

Reviewer #1 (Remarks to the Author):

The authors have addressed my concerns.

Reviewer #2 (Remarks to the Author):

The authors had addressed and answered my questions.

Reviewer #3 (Remarks to the Author):

Wu and colleagues study the effect of APPL1 deficiency on NLRP3 activation both in vitro and in vivo. The document a role for endosomal Rab5 regulating mitophagy, without which, excess mtDNA promotes NLRP3 activation. The revised manuscript appears to address some points, but one result in particular is a major issue.

Major point:

Caspase-11 activation does not result in IL-1b cleavage directly, only indirectly via NLRP3 (<https://pubmed.ncbi.nlm.nih.gov/26173909/>). Therefore, the new Supp Fig 1G where there is no difference in IL-1b for transfected LPS to activate Casp-11/NLRP3 argues against the central tenet for the paper.

Minor point:

The effect of the cGAS inhibitor to reduce the difference in APPL1 ko cells after NLRP3 stimulation suggests that this pathway is involved, but seems to contradict the effect of STING siRNA experiments where there is no difference.

Point-by-point replies to the reviewers

(NCOMMS-20-26496-T, by Wu KKL et al, entitled “An early endosomal-dependent mitophagy mediated by APPL1-Rab5 axis restricts NLRP3 inflammasome activation in macrophages”)

First of all, we would like to express our sincere thanks to the editor and reviewers for their constructive comments. We have now revised our manuscript according to your comments. Our point-by-point responses are listed below.

Reviewer #3 (Remarks to the Author)

Wu and colleagues study the effect of APPL1 deficiency on NLRP3 activation both in vitro and in vivo. The document a role for endosomal Rab5 regulating mitophagy, without which, excess mtDNA promotes NLRP3 activation. The revised manuscript appears to address some points, but one result in particular is a major issue.

Major point: Caspase-11 activation does not result in IL-1b cleavage directly, only indirectly via NLRP3 (<https://pubmed.ncbi.nlm.nih.gov/26173909/>). Therefore, the new Supp Fig 1G where there is no difference in IL-1b for transfected LPS to activate Casp-11/NLRP3 argues against the central tenet for the paper.

Response: The central tenet of our manuscript is that APPL1 deficiency induced-defective mitophagy and consequent production of mitochondrial danger signal is the underlying cause of NLRP3 inflammasome in macrophages. However, whether LPS transfection affects mitophagy and/or elicits mitochondria danger signals remain uncertain in the current state. The major conclusion from the study by Sebastian Ruhl et al (<https://pubmed.ncbi.nlm.nih.gov/26173909/>) is that activation of caspase-11 by LPS transfection induces potassium efflux, which leads to NLRP3 activation and IL-1 β secretion in macrophages^[1]. Furthermore, the authors showed that LPS transfection appears to induce mitochondrial ROS (mtROS) production (Figure 2B and 2D, and extracted diagram below).

Figure 2. Involvement of mitochondria in noncanonical NLRP3 activation. (A) LDH and IL-1 β release from PAM₃CSK₄-primed WT BMDMs transfected with LPS for 4 h or stimulated with nigericin for 1 h in Opti-MEM with or without 10 mM NAC. (B) Flow cytometry analysis of mitochondrial ROS levels in PAM₃CSK₄-primed WT BMDMs transfected with LPS for 4 h or transfected with deoxy-adenosine:deoxy-thymidine (0.3 μ g/2.5 \times 10⁴ cells) to stimulate caspase-1-induced mitochondrial ROS via the AIM2 inflammasome for 3 h in Opti-MEM with or without 10 mM NAC. Live cells were analyzed for MitoSOX staining by flow cytometry (as shown in F). Bars represent the fold change in MFI compared to nonstimulated controls. (C and D) LDH and IL-1 β release and flow cytometry analysis of mitochondrial ROS levels (as described in F) in PAM₃CSK₄-primed WT BMDMs transfected with LPS for 4 h or stimulated with nigericin for 0.5 h in Opti-MEM with or without 1 mM MitoTEMPO. (E) LDH and IL-1 β release from PAM₃CSK₄-primed WT BMDMs infected with log-phase *S. typhimurium* (MOI 25) for 1 h in Opti-MEM with or without 1 mM MitoTEMPO (x -axis indicates hours of MitoTEMPO incubation). (F) Staining procedure and gating scheme for MitoSOX analysis. ND, not detectable. All data show average (\pm SD) of triplicate wells representative of at least two independent experiments. * p < 0.05, Student's t -test.

<https://pubmed.ncbi.nlm.nih.gov/26173909/>

Reply to the Reviewer #3

Since the authors did not indicate the statistical comparison between not stimulated and LPS transfection, we therefore do not know whether LPS transfection significantly upregulates mtROS production in macrophages (highlighted with red square in the above Figure). The authors also attempted to investigate whether the mtROS (the potential mitochondrial danger signal) mediates LPS transfection-induced NLRP3 activation using N-Acetyl-Cysteine (NAC) and MitoTempo treatment. Unfortunately, given the off-target effect of the ROS scavengers, the authors were unable to conclude whether mitochondria dysfunctions/mtROS participate in caspase-11-induced NLRP3 activation.

Sebastian Ruhl et al stated their conclusion as follow (p.2930 of the paper, the last paragraph):

“We concluded that due to off-target effects ROS scavengers cannot be used to study the involvement of mitochondria in caspase-11-driven NLRP3 activation, yet we cannot exclude a possible involvement of mitochondrial ROS or other factors of mitochondrial physiology such as the release of oxidized mitochondrial DNA or the perturbation of the mitochondrial membrane potential in caspase-11-induced NLRP3 activation.”

In addition to the article listed by the Reviewer 3, a recent publication showed that LPS transfection triggers mitochondrial DNA release to the cytosol via induction of the pore-forming protein Gasdermin D, which in turn activates cGAS and then suppresses proliferation in endothelial cells^[2]. However, whether this detrimental effect of LPS transfection on mitochondria also occurs in macrophages and its relevance to NLRP3 activation remains uncertain. Apart from the above two articles, to our best knowledge, we failed to find any other studies showing the effect on LPS transfection on mitophagy/production of mitochondrial danger signals. Therefore, the effect of LPS transfection on mitochondria in macrophage requires further investigation. However, this is out of scope of our current manuscript.

As mentioned above, the central tenet of our manuscript is that defective mitophagy induced by aberrant APPL1-Rab5 axis leads to excessive accumulation of damaged mitochondria, which in turn produces mitochondrial danger signals (such as mtROS and hence oxidized cytosolic mtDNA), leading to NLRP3 inflammasome activation. Unlike LPS transfection, the treatment with ATP, nigericin, MSU, FCCP, rotenone, antimycin or palmitic acid (Figure 1, Supplementary Figure 1, 3 and 13) in LPS-primed macrophage are well-known to induce mitochondrial damages and NLRP3 inflammasome activation^[3-5]. Therefore, we believe that our findings firmly support the central tenet of our manuscript.

Minor point:

The effect of the cGAS inhibitor to reduce the difference in APPL1 ko cells after NLRP3 stimulation suggests that this pathway is involved, but seems to contradict the effect of STING siRNA experiments where there is no difference.

Response: In Supplementary Fig. 4E, we found that APPL1 KO BMDM exhibited significant elevation of IL-1 β secretion when compared to WT controls in the presence or absence of the

cGAS inhibitor (RU521), indicating that cGAS plays a minimal role, if any, in APPL1 deficiency-induced NLRP3 inflammasome activation.

In contrast to RU521 treatment, inhibition of ROS, removal of mitochondrial DNA or inhibition of oxidized mtDNA to NLRP3 almost completely abrogated the potentiating effect of APPL1 deficiency on NLRP3 inflammasome activation (Figure 3). Please also refer our description in Result Section, and we extract here for your reference:

“The excessive accumulation of cytosolic mtDNA might induce inflammasome overactivation in APPL1 KO BMDM, via AIM2 or cGAS-STING-dependent pathway³²⁻²⁴. To test the above possibility, we employed siRNA to knockdown AIM2 or STING expression in APPL1 KO BMDM. As expected, downregulation of AIM2 or STING abolished Poly (dA:dT)-induced IL-1 β production in both APPL1 KO BMDM or WT BMDM, yet it had minimal effect on APPL1 deficiency-induced augmented NLRP3 inflammasome activation under LPS+ATP or LPS+FCCP condition (Supplementary Fig. 4A-E). Surprisingly, pharmacological inhibition of cGAS using RU521 partially attenuated APPL1 deficiency-induced NLRP3 inflammasome activation (Supplementary Fig. 4F). However, comparing with the inhibitory effects observed in MitoTEMPO, ethidium bromide and 8-OH-dG experiments (Fig. 3), IL-1 β production remained significantly higher in APPL1 KO BMDM than that in WT BMDM under RU521 treatment condition. These data collectively suggest that AIM2 or cGAS/STING might not be the major mediator of APPL1 deficiency-induced NLRP3 inflammasome activation in BMDM.”

Since its discovery in 2017, RU521 has been recently used in few studies in the investigation of the cGAS function/signalling [6-8]. Since RU521 is a chemical inhibitor, its off-target effect cannot be fully excluded. On the other hand, we are willing to repeat the experiment using siRNA against cGAS and determine whether silencing of cGAS abrogates the promoting effect of APPL1 deficiency on NLRP3 activation, if the Editor/Reviewer feels this is a major concern.

References:

1. Ruhl, S. and P. Broz, *Caspase-11 activates a canonical NLRP3 inflammasome by promoting K(+) efflux*. Eur J Immunol, 2015. **45**(10): p. 2927-36.
2. Huang, L.S., et al., *mtDNA Activates cGAS Signaling and Suppresses the YAP-Mediated Endothelial Cell Proliferation Program to Promote Inflammatory Injury*. Immunity, 2020. **52**(3): p. 475-486 e5.
3. Zhou, R., et al., *A role for mitochondria in NLRP3 inflammasome activation*. Nature, 2011. **469**(7329): p. 221-5.
4. Wen, H., et al., *Fatty acid-induced NLRP3-ASC inflammasome activation interferes with insulin signaling*. Nat Immunol, 2011. **12**(5): p. 408-15.
5. Nakahira, K., et al., *Autophagy proteins regulate innate immune responses by inhibiting the release of mitochondrial DNA mediated by the NALP3 inflammasome*. Nat Immunol, 2011. **12**(3): p. 222-30.
6. Ma, C., et al., *Gasdermin D in macrophages restrains colitis by controlling cGAS-mediated inflammation*. Sci Adv, 2020. **6**(21): p. eaaz6717.
7. Vincent, J., et al., *Small molecule inhibition of cGAS reduces interferon expression in primary macrophages from autoimmune mice*. Nat Commun, 2017. **8**(1): p. 750.

Reply to the Reviewer #3

8. Wisner, C., et al., *Small molecule inhibition of human cGAS reduces total cGAMP output and cytokine expression in cells*. Sci Rep, 2020. **10**(1): p. 7604.